# HIPODE: Enhancing Offline Reinforcement Learning with High-Return Synthetic Data from a Policy-Decoupled Approach

## Abstract

Offline reinforcement learning (Offline RL) has gained attention as a means of training reinforcement learning models using pre-collected static data. To address the issue of limited data and improve downstream Offline RL performance, recent efforts have focused on broadening dataset coverage through data augmentation techniques. However, most of these methods are tied to a specific policy (policy-dependent), restricting the generated data to supporting only a specific downstream Offline RL policy. Moreover, the return (quality) of synthetic data is often not well-controlled, which limits the potential for further improving the downstream policy. To tackle these issues, we propose **HI**gh-return **PO**licy-**DE**coupled (HIPODE), a novel data augmentation method for Offline RL. On the one hand, HIPODE generates high-return synthetic data by selecting states near the dataset distribution with potentially high value among candidate states using the negative sampling technique. On the other hand, HIPODE is policy-decoupled, thus can be used as a common plug-in method to support diverse downstream Offline RL processes. We conduct experiments on the widely studied TD3BC, CQL and IQL algorithms, and the results show that HIPODE outperforms or has competitive results to the state-of-the-art policy-decoupled data augmentation method and most prevalent model-based Offline RL methods on D4RL benchmarks.

## 1 Introduction

Reinforcement learning (RL) (Sutton & Barto, 2018; Precup et al., 2001; Sutton, 1991) has achieved remarkable success in various applications, including game playing (Mnih et al., 2013), robotics (Singh et al., 2022), finance (Charpentier et al., 2021), and healthcare (Liu et al., 2020a). Nevertheless, traditional RL algorithms necessitate real-time interaction with the environment. In many real-world scenarios, such as autonomous driving (Kiran et al., 2021) and medical treatment (Liu et al., 2020a), online learning is not feasible due to the high cost of failures, and collecting new data is often expensive or even dangerous (Prudencio et al., 2023). Consequently, Offline RL (Offline RL) (Lange et al., 2012; Levine et al., 2020) has garnered significant attention in recent years as it aims to learn from a dataset of previously collected experiences without further interaction with the environment.

In the offline setting, prior off-policy RL methods are known to fail on fixed offline datasets (Haarnoja et al., 2018; Fujimoto et al., 2018), even on expert demonstrations (Fujimoto et al., 2019). The main reason of this could be the limited coverage of offline data. This can cause the policy visiting states that are out of the distribution (OOD) of the dataset, and suffer from the extrapolation error on these states (Fujimoto et al., 2019; Kumar et al., 2019). To alleviate extrapolation errors, most Offline RL researches attempt to avoid out-of-distribution states or actions in actor-critic iterations, focusing on policy constraint (Fujimoto et al., 2019; Wu et al., 2019; Liu et al., 2020b; Fujimoto & Gu, 2021), support constraint (Kostrikov et al., 2022; Kumar et al., 2019), value regularization (Kumar et al.; Ma et al., 2021b;a; Kumar et al., 2021; Kostrikov et al., 2021; An et al., 2021), and others. However, these approaches face the problem of the loss of generalization capability (Lyu et al., 2022).

Different from mitigating the extrapolation error in actor-critic iterations, data augmentation has been applied in Offline RL recently to expand the coverage of the dataset. The simplest approach is to add noise to the original dataset to obtain augmented data (Sinha et al., 2022; Weissenbacher et al., 2022),

which could result in inaccurate dynamics transition that may not match the real environment. In contrast, dynamics models used in model-based RL can augment the dataset by rolling out synthetic samples. Inspired by this, existing works use the forward or backward dynamics models (Yu et al., 2021; 2020; Kidambi et al., 2020; Lyu et al., 2022; Wang et al., 2021; 2022; Lu et al., 2022; Guo et al., 2022; Rigter et al., 2022; Fu et al.) to generate synthetic data and incorporate them into the policy training process. However, most of these methods are policy-dependent since they have to explicitly deal with unreliable data derived from inaccurate models to adapt to the downstream policy, thus limiting their data's application to augment other Offline RL algorithms. Among them, (Wang et al., 2021; Lyu et al., 2022) achieve policy-decoupled data augmentation. However, these methods lack explicit constraints to ensure the return of the generated data, making the underlying mechanism by which they work unclear and limiting the potential of further improvement to the downstream policy. On the other hand, previous research has indicated that RL algorithms derive advantages from highly diverse data (Kumar et al., 2022; Yarats et al., 2022). However, these studies have been conducted with the collection of accurate and diverse ground-truth data from the environment, which is substantially different from our setting.

To overcome the above-mentioned issues, we investigate the data augmentation method that is not dependent on the downstream Offline RL policy, while also ensuring the high-return of the generated data. We first empirically analyze that high-return data is beneficial for enhancing Offline RL performance. Then, we propose the **HI**gh-return **PO**licy-**DE**coupled (HIPODE) data augmentation approach.

Specifically, we empirically show that high-return data can be more efficient for enhancing Offline RL performance than high diversity data, by comparing data from two types of augmentation policies: a noisy policy and an offline RL policy, and using the true environment as the data generator to eliminate the influence of model inaccuracy. Our findings show that data from a noisy policy rarely beneficial for downstream offline policy learning algorithms, and may even be harmful when the noise is too violent. In contrast, high-return data can enhance the downstream policy.

We present the outline of our policy-decoupled data augmentation process for Offline RL in Fig.1, which involves using specific augmentation policy to generate synthetic datasets based on the original dataset. These synthetic datasets are then used to expand the training data for any downstream Offline RL algorithm. In policy-dependent methods, the augmentation policy is related to the downstream policy (green arrow), which we try to break (the red cross). Throughout the process in Fig. 1, our key insight is to generate high-return synthetic augmented data while ensuring authenticity (i.e. the proximity level between the synthetic data and the real

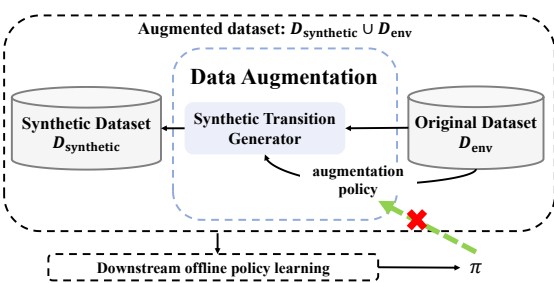

Figure 1: Outline of our data augmentation for Offline RL.

data) as much as possible in such a policy-decoupled way. To summarize, the contributions of this paper are:

- We investigate the impact of different types of augmented data on downstream Offline RL algorithms. Our findings indicate that high-return data, as opposed to noisy data with high diversity, benefits downstream offline policy learning performance more.

- We propose a novel policy-decoupled data augmentation method HIPODE for Offline RL. HIPODE serves as a common plugin that can augment high-return synthetic data for any Offline RL algorithm, and is decoupled with downstream offline policy learning process.

- We evaluate HIPODE on D4RL benchmarks and it significantly improves several widely used model-free Offline RL baselines. Furthermore, HIPODE outperforms state-of-the-art (SOTA) policy-decoupled data augmentation approaches for Offline RL.

## 2 RELATED WORK

**Data augmentation in Offline RL.** To address the challenge of limited data in Offline RL, various methods have been proposed to generate more sufficient data. Most of these approaches are policy-dependent, meaning they generate data based on the current policy and use it to refine the training of the same policy (Wang et al., 2022). These policy-dependent data augmentation methods can be divided into two categories. The first category seeks to generate pessimistic synthetic data that would be pes-

Table 1: Results of downstream Offline RL algorithm using different types of augmented data. Org denotes using the original dataset; Org + Div $\sigma$ and Org + Return denote using high-diversity or high-return augmented data; -r and -m-r denotes -random-v0 and -medium-replay-v0.

| Task Name | Type | TD3BC | CQL | IQL |
|---|---|---|---|---|
| halfcheetah-r | Org | $12.8 \pm 1.9$ | $17.0 \pm 9.3$ | $16.5 \pm 0.6$ |
| | Org + Div 0.01 | $12.1 \pm 0.6$ | $2.0 \pm 0.2$ | $11.2 \pm 4.0$ |
| | Org + Div 0.1 | $12.0 \pm 0.5$ | $16.1 \pm 5.5$ | $14.3 \pm 3.0$ |
| | Org + Div 1.0 | $9.2 \pm 0.7$ | $3.0 \pm 0.4$ | $12.6 \pm 6.2$ |
| | Org + Return | $\mathbf{25.8} \pm 0.0$ | $\mathbf{23.8} \pm 0.3$ | $\mathbf{21.2} \pm 4.7$ |
| halfcheetah-m-r | Org | $43.3 \pm 0.6$ | $42.5 \pm 0.9$ | $41.1 \pm 1.1$ |
| | Org + Div 0.01 | $44.6 \pm 0.6$ | $38.4 \pm 6.7$ | $43.3 \pm 0.3$ |
| | Org + Div 0.1 | $44.3 \pm 0.3$ | $1.8 \pm 0.1$ | $44.0 \pm 0.2$ |
| | Org + Div 1.0 | $41.8 \pm 0.7$ | $25.6 \pm 19.2$ | $40.1 \pm 0.3$ |
| | Org + Return | $\mathbf{46.8} \pm 0.1$ | $\mathbf{52.6} \pm 0.1$ | $\mathbf{44.6} \pm 0.2$ |

simistic enough if it is OOD, thus expand the dataset's coverage while mitigating the extrapolation error caused by such OOD data. The literature (Yu et al., 2020; Kidambi et al., 2020) rely on the disagreement of dynamics ensembles or Q ensembles to construct a pessimistic MDP, and (Yu et al., 2021; Rigter et al., 2022; Guo et al., 2022) achieve the underestimation of synthetic data by unrolling the current policy in the model. The second category does not explicitly pursue the underestimation of synthetic data. Among them, (Fu et al.) generates and selected synthetic data with low model disagreement, and BooT (Wang et al., 2022) augments TT (Janner et al., 2021) with the synthetic data generated by itself. Besides, S4RL (Sinha et al., 2022) and KFC (Weissenbacher et al., 2022) add noise in a local area of states to smooth the critic.

An obvious drawback of these aforementioned approaches, which are all policy-dependent, is that the generated data is closely related on the policy itself, causing that applying the generated data directly to the learning process of other policies is not guaranteed to perform well (Wang et al., 2022). To overcome this limitation, recent studies have explored policy-decoupled data augmentation techniques, which is the focus of this paper. Bi-directional rolling proposed in (Lyu et al., 2022) induce the double-check mechanism into offline data augmentation ensure that the generated data is within the distribution and avoids inauthentic samples. In (Wang et al., 2021), a reverse dynamics model is proposed for Offline RL. These two methods use the behavioural policy as the augmentation policy, thus achieving policy-decoupled data augmentation. However, these two methods only consider the reliability of the synthetic data and neglect the return of generated data, which may limit the performance. Another recent work SynthER (Lu et al., 2023) focuses on enhancing overall performance by scaling up downstream networks and utilizing diffusion model data augmentation to address overfitting issues, resulting in significant overall improvements. MOCODA (Pitis et al., 2022) and GuDA (Corrado et al., 2023) propose two different frameworks for controlled distribution of OOD data augmentation but both require additional given expert knowledge of the specific task environment.

Different from the above methods, HIPODE's focus is on strengthening offline RL algorithms with minimal cost, particularly aiming to improve performance when downstream networks remain unchanged and relatively small by generating high-return synthetic data.

**Model-free Offline RL.** Disregarding data augmentation, the model-free Offline RL algorithm investigates how to constrain the policy to approach the behavioral policy or support in static offline datasets. Existing methods implement this by policy constraint (Fujimoto et al., 2019; Wu et al., 2019; Liu et al., 2020b; Fujimoto & Gu, 2021), support constraint (Kostrikov et al., 2022; Kumar et al., 2019), value regularization (Kumar et al.; Ma et al., 2021b;a; Kumar et al., 2021; Kostrikov et al., 2021; An et al., 2021), and others. Among them, we choose widely-used TD3BC (Fujimoto & Gu, 2021) and CQL (Kumar et al.) to be the downstream policy learning algorithm to evaluate different data augmentation methods.

## 3 WHAT KIND OF DATA IS MORE APPROPRIATE FOR OFFLINE DATA AUGMENTATION?

As mentioned before, data augmentation methods to a specific dataset in Offline RL often neglect the data return (quality). However, high-return data are regarded as beneficial for learning (Fu et al., 2020). Accordingly, we pose the question of whether the generation of high-return data is also beneficial for Offline RL policies when compared to high-diversity data. We primarily investigate this question in this section.

To fairly investigate the effect of high-diversity data and high-return data on the downstream Offline algorithm, we generate two types of data from real environments, instead of generating from other data augmentation techniques, to prevent potential bias due to inauthentic data impacting our findings. Although we use the environment, we limit the data generated in this section to only those that are not far away from the original dataset, to more closely match the offline setting.

Concretely, we choose the following two types of augmentation policies: (1) **Policy of high diversity**, where random noise with different scales is added to the behavioral policy. Formally, $\pi_{\text{noise}} := \mathcal{N}(a, \sigma I)$, s.t., $a \sim \pi_\beta$, where $\pi_\beta$ denotes the behavioural policy, $\mathcal{N}$ denotes the Gaussian distribution and I denotes a identity matrix. The dataset after augmentation by this policy exhibits higher **diversity** compared to the original dataset. We refer to this type of method as 'Diversity $\sigma$', where $\sigma$ belongs to 0.01, 0.1, 1.0. (2) **Policy of high-return**, a well-trained Offline policy, to ensure the action **return**, derived from the Offline policy that is similar to or higher than that of the actions in the dataset overall. Meanwhile, the generated data is also ensured to be close to the dataset. We refer this as 'Return'. We provide the experimental details in Appendix B.2.

The augmented data and the original data are together used to train the downstream Offline algorithms. Normalized score reported in Table 1 shows that using Offline policy to augment data can always benefit down stream offline policy learning performance while using random noise policy may not. We further visualize the distribution of the original dataset, the noise-policy-augmented data, and the high-return-policy-augmented data through t-Distributed Stochastic Neighbor Embedding (t-SNE) (Hinton & Roweis, 2002) in Fig.2. As we can see from it, compared with the distribution of the original data, the distribution of the noise-policy-augmented data is similar to the original dataset's while the distribution of the high-return-policy-augmented data is relatively concentrated in several clusters. In addition, the high-return-policy-augmented data indeed has higher rewards in a single time-step, as the color

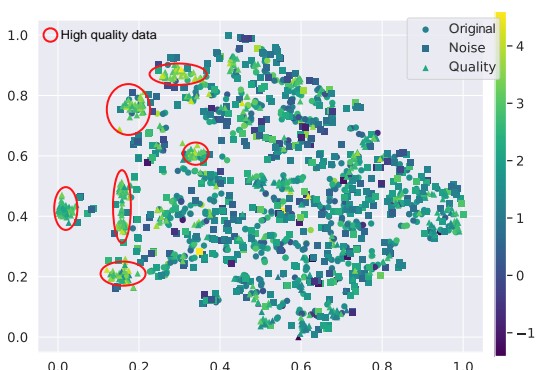

Figure 2: Action distributions of the original dataset, noise-policy-augmented data and high-return-policy-augmented data. Brighter color indicates higher reward in a single time-step.

of the most triangle points are brighter. Based on these observations, we present the following takeaway:

> **Takeaway**: in the case that the augmented data is completely realistic, data with higher return may be more beneficial than that of more diversity in improving downstream Offline algorithm.

## 4 METHOD

According to the takeaway above, we propose HIPODE to generate augmented data that maximizes its return, while maintaining as much authenticity as possible in a policy-decoupled way. We illustrate HIPODE in Fig.3. Specifically, given any state $s$, we first generate several candidate next states $\tilde{S}'_{\text{cand}} = \{\tilde{s}'_1, ..., \tilde{s}'_n\}$ (Step 1 in Fig.3). Then we select the one with the highest value as $\tilde{s}'$ (Step 2).

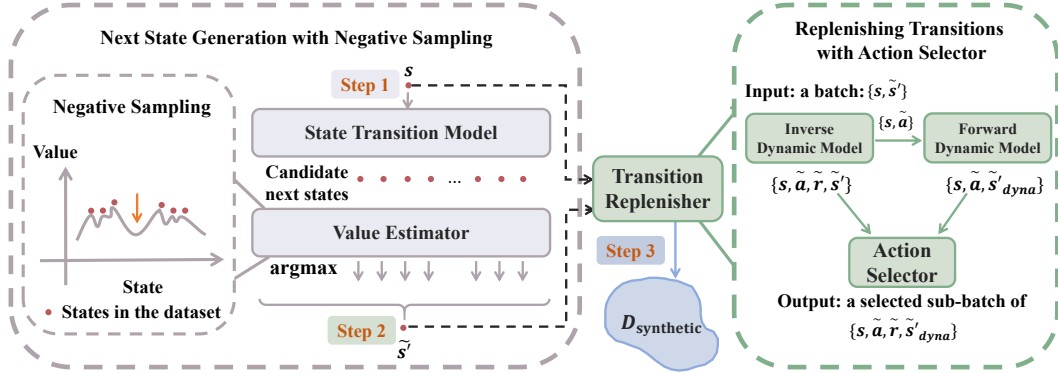

Figure 3: Illustration of HIPODE.

Finally, given $s$ and $\tilde{s}'$, the action $\tilde{a}$ and the reward $\tilde{r}$ are produced using generative models (Step 3), thus generating a transition $\{s, \tilde{a}, \tilde{r}, \tilde{s}'\}$. In the following, we introduce Step 1 and 2 in Section 4.1 and Step 3 in Section 4.2.

## 4.1 Next State Generation with Negative Sampling

Given a state $s$, we generate the next state through a state transition model, and filter the high-return data for our purpose. In the following, we describe these two steps in detail.

**The forward state transition model.** We first train a state transition model $\tilde{p}_\psi(s'|s)$ to generate candidate next states. To guarantee the authenticity of generated next state, we model the state transition within the dataset with a conditional variational auto-encoder (CVAE) following (Zhang et al., 2022), to ensure the generated next states are near the distribution of the dataset. Specifically, CVAE consists of an encoder and a decoder: the encoder takes the current state and the next state as input and manages to output an latent variable $z$ under the Gaussian distribution; the decoder takes $z$ and the current state as input and manages to map the latent variable $z$ to the desired space. We denote the encoder as $E_\psi(s, s')$ and the decoder as $D_\psi(s, z)$. The state transition model is then trained by maximizing its variational lower bound, which is equivalent to minimizing the following loss:

$$\mathcal{L}(\psi) = \mathbb{E}_{(s,s')\sim\mathcal{D}_{\text{env}}, z\sim E_\psi(s,s')}[(s' - D_\psi(s, z))^2 + D_{\text{KL}}(E_\psi(s, s')\|\mathcal{N}(0, \text{I}))]. \quad (1)$$

where I represents an identity matrix and $\mathcal{D}_{\text{env}}$ represents the original dataset. The first term of RHS of Eq.1 represents the reconstruction loss where the approximated next state is decoded from $z$, given the current state. The second term of RHS represents the KL distance between the distribution of $z$ and the Gaussian distribution so that a sampled $z$ from a Gaussian distribution can be decoded to the desired state space when generating. Thus, given a state $s$, $n$ candidate next states $\tilde{S}'_{\text{cand}} = \{\tilde{s}'_1, ..., \tilde{s}'_n\}$, s.t., $\tilde{s}'_i \sim D_\psi(\tilde{s}'_i|s, z)$ are sampled.

**Value Approximation with Negative Sampling.** To filter out the generated next states and form synthetic transitions, a value approximator is trained using SARSA-style updating to predict the value of different states. Since the generated next states may not be present in the dataset, the negative sampling technique (Luo et al., 2020) is employed to avoid overestimation of states outside the dataset. Specifically, for states within the dataset, standard TD-learning is performed as demonstrated in Eq.2:

$$\mathcal{L}^{\text{td}}_\theta(s) = \mathbb{E}_{(s,r,s')\sim\mathcal{D}_{\text{env}}}[r + \gamma V_{\bar{\theta}}(s') - V_\theta(s)]^2, \quad (2)$$

where $V_{\bar{\theta}}$ is the target value function and $\gamma$ is the discount factor. Furthermore, to conservatively estimate the value of states outside the dataset distribution, we sample states around the dataset states by adding Gaussian noise and evaluate the L2 distances between the sampled noisy states and the original states. The greater the distance between the sampled state and the original state, the more severe the penalties imposed on the sampled state, as shown in Eq.3:

$$\mathcal{L}^{\text{ns}}_\theta(s) = \mathbb{E}_{s\sim\mathcal{N}(s_{\text{d}},\sigma\text{I}),(s_{\text{d}},r,s')\sim\mathcal{D}_{\text{env}}}[r + \gamma V_{\bar{\theta}}(s') - \alpha\|s - s_{\text{d}}\| - V_\theta(s)]^2, \quad (3)$$

where $s_{\mathrm{d}}$ denotes the state sampled from the original dataset and $\alpha$ denotes the penalty weight. Thus, the optimization objective of the value approximator to minimize is represented by Eq.4:

$$\mathcal{L}(\theta) = \mathcal{L}(\theta)^{\mathrm{td}} + \mathcal{L}(\theta)^{\mathrm{ns}}. \tag{4}$$

After training, all the candidate next states $S'_{\mathrm{cand}}$ are input into the value approximator to obtain their values. Then the candidate next state with the highest value estimation is selected, formally $\tilde{s}' = \mathrm{argmax}_{\tilde{s}'_{\mathrm{cand}} \in \tilde{S}'_{\mathrm{cand}}} V(\tilde{s}'_{\mathrm{cand}})$. Intuitively, a state can be selected in two cases:

- States within the dataset. This is because other candidate states that not in the dataset are severely underestimated. In this case, the selected state can be considered reliable.

- States with high true value near the dataset distribution. Since its estimated value is significantly penalized during training, there is a high probability that a selected state close to the distribution has a high true value.

Therefore, by filtering the candidate next states generated by the state transition model using the value approximator, we can obtain the augmented next state with similar or higher return than that in the datasets while maintaining as much authenticity as possible.

## 4.2 REPLENISHING TRANSITIONS WITH ACTION SELECTOR

Based on the selected high-return next state, in this section, we aim to generate an authentic action that can lead the current state to the generated next state. Specifically, an inverse model $M_{\mathrm{inv}} = \tilde{p}_\epsilon(a|s, s')$ is trained to generate actions conditioned on $s$ and the selected $\tilde{s}'$. Similar to the state transition model, we also use a CVAE for generating actions. We denote the encoder as $E_\epsilon(a, s, s')$ and the decoder as $D_\epsilon(s, s', z)$. The inverse model is then trained by maximizing its variational lower bound, which is equivalent to minimizing the following loss shown as Eq.5:

$$\mathcal{L}(\epsilon) = \mathbb{E}_{(a,s,s') \sim \mathcal{D}_{\mathrm{env}}, z \sim E_\epsilon(a,s,s')}[(a - D_\epsilon(s, s', z))^2 + D_{\mathrm{KL}}(E_\epsilon(a, s, s') \| \mathcal{N}(0, \mathrm{I}))]. \tag{5}$$

Besides, rewards are generated the same way as actions, using another model with encoder $E_\zeta(r, s, s')$ and decoder $D_\zeta(s, s', z)$.

Although the generated state have high return and authenticity as described in Section 4.1, the action generated by the inverse dynamics model may be inauthentic, i.e. the generated action can not lead to the selected next state. Therefore, a filtering mechanism is imposed on actions for their reliability. We further draw on a forward dynamics model $M_{\mathrm{for\_dyna}} = \tilde{p}_w(\tilde{s}'_{\mathrm{dyna}}|s, a)$ representing the probability of the next state given the current state and action. The dynamics model is optimized by maximizing the log-likelihood of the static dataset, formally shown in Eq.6:

$$\mathcal{L}(w) = \mathbb{E}_{(s,a,s') \sim \mathcal{D}_{\mathrm{env}}}[-\log \tilde{p}_w(s'|s, a)]. \tag{6}$$

Combining the forward dynamics model $M_{\mathrm{for\_dyna}}$ and the inverse dynamics model $M_{\mathrm{inv}}$, an action is assumed to be reliable to lead the selected state $\tilde{s}'$ when the distance between the selected state $\tilde{s}'$ and the forward-predicted state $\tilde{s}'_{\mathrm{dyna}} = M_{\mathrm{for\_dyna}}(s, \tilde{a})$ is small enough. In practice, instead of setting a threshold for measuring the distance between $\tilde{s}'$ and $\tilde{s}'_{\mathrm{dyna}}$, we pick up in a batch $\lambda$-portion of the generated data with the lowest $\|\tilde{s}'_{\mathrm{dyna}} - \tilde{s}'\|$ values and consider them the most reliable subset of the batch. Then this subset of data is used as the final augmented data.

Although HIPODE consists of several modules, it is highly integrated. Thus it's a common plug-in method, which can be simply combined to diverse downstream Offline RL algorithms by only merging the synthetic data of HIPODE and the original data. For the summary and the pseudocode of HIPODE, please refer to Appendix A.

## 5 EXPERIMENTS

In this section, we evaluate HIPODE based on two representative and widely-used offline policy learning algorithm: TD3BC (Fujimoto & Gu, 2021) and CQL (Kumar et al.).We aim to answer these questions:

*Q1*: Can our proposed algorithm HIPODE improve existing Offline algorithms and exhibit consistent superiority in comparison to other data augmentation technique?

Table 2: Normalized average score and standard deviation over at least 3 seeds (5 seeds for HIPODE+CQL) of HIPODE based on downstream Offline RL algorithms (CQL and TD3BC) and baseline performance. In the table, -m-e, -m-r, -m, -r, -e denote -medium-expert, -medium-replay, -medium, -random, -expert respectively. The numbers in blue represent the rate of enhancement from HIPODE. The best result in each row is **bolded**.

| Task Name | CQL + HIPODE | CQL + CABI | CQL | TD3BC + HIPODE | TD3BC + CABI | TD3BC | COMBO | MOPO |
|---|---|---|---|---|---|---|---|---|
| halfcheetah-m-e | $101.8 \pm 3.2$ | $101.0\pm1.2$ | $94.8\pm3.9$ | **$102.6$**$\pm2.2$ | $94.6\pm8.5$ | $98.0\pm3.1$ | 38.7 | 63.3 |
| hopper-m-e | $112.1 \pm 0.1$ | $112.0\pm0.1$ | $111.9\pm0.1$ | **$112.4$**$\pm0.3$ | $102.8\pm16.0$ | $112.0\pm0.1$ | 75.1 | 23.7 |
| walker2d-m-e | $91.7 \pm 7.1$ | $92.3\pm13.8$ | $70.3\pm5.7$ | $105.3 \pm 4.0$ | $99.6\pm6.4$ | **$105.4$**$\pm3.9$ | 2.3 | 44.6 |
| halfcheetah-m-r | $44.8\pm0.4$ | $42.8\pm2.0$ | $42.5\pm0.9$ | $44.0 \pm 0.7$ | $43.4\pm0.5$ | $43.3\pm0.6$ | 46.9 | **53.1** |
| hopper-m-r | $33.5 \pm 2.9$ | $29.7\pm2.7$ | $28.2\pm1.2$ | $36.2\pm0.8$ | $32.7\pm2.3$ | $32.7\pm0.5$ | 19.7 | **67.5** |
| walker2d-m-r | $21.7 \pm 6.5$ | $12.7\pm6.7$ | $5.1\pm3.6$ | $36.4 \pm 11.1$ | $37.7\pm15.4$ | $19.6\pm8.6$ | 19.5 | **39.0** |
| halfcheetah-m | $39.3 \pm 0.2$ | $36.7\pm0.2$ | $39.2\pm0.4$ | **$43.7$**$\pm0.5$ | $43.0\pm0.3$ | $43.7\pm0.4$ | 27.4 | 42.3 |
| hopper-m | $30.4 \pm 0.9$ | $30.5\pm0.3$ | $30.3\pm0.7$ | **$99.9$**$\pm0.3$ | $99.7\pm0.2$ | **$99.9$**$\pm0.9$ | 71.6 | 28.0 |
| walker2d-m | $79.9 \pm 3.2$ | $46.8\pm16.1$ | $66.9\pm9.3$ | $80.1 \pm 0.7$ | **$80.3$**$\pm2.0$ | $79.7\pm1.8$ | 71.8 | 17.8 |
| halfcheetah-r | $22.9\pm2.7$ | $2.8\pm0.9$ | $17.0\pm5.7$ | $15.5 \pm 0.9$ | $12.5\pm1.5$ | $12.8\pm1.9$ | 5.5 | **35.4** |
| hopper-r | $10.4 \pm 0.1$ | $10.2\pm0.1$ | $10.4\pm0.1$ | $10.9\pm0.2$ | $10.9\pm0.0$ | $10.9\pm0.2$ | 7.5 | **11.7** |
| walker2d-r | **$16.5$**$\pm10.5$ | $-0.1\pm0.0$ | $1.7\pm0.9$ | $6.6 \pm 1.0$ | $3.0\pm0.7$ | $0.37\pm0.1$ | 1.6 | 13.6 |
| **Total** | $605.0 \uparrow16.7\%$ | $517.4$ | $518.3$ | **$693.6 \uparrow5.4\%$** | $660.2$ | $658.3$ | 387.5 | 440.0 |
| halfcheetah-e | $108.5 \pm 1.4$ | $109.1\pm0.3$ | **$109.8$**$\pm0.3$ | $106.5 \pm 0.5$ | $106.3\pm1.0$ | $107.0\pm0.9$ | 44.2 | 102.1 |
| hopper-e | $112.2 \pm 0.2$ | **$112.3$**$\pm0.2$ | $112.0\pm0.1$ | **$112.3$**$\pm0.5$ | $110.8\pm0.3$ | $107.3\pm8.7$ | **112.3** | 0.7 |
| walker2d-e | **$109.9$**$\pm3.6$ | $98.7\pm19.6$ | $104.7\pm5.0$ | $106.6 \pm 5.2$ | $102.5\pm5.2$ | $98.7\pm6.0$ | 37.3 | 2.1 |
| **Total** | $935.6$ | $837.5$ | $844.9$ | **$1019.0$** | $979.8$ | $971.5$ | 581.3 | 544.9 |
| **Avg** | $62.4 \uparrow10.8\%$ | $55.8$ | $56.3$ | **$67.9 \uparrow4.8\%$** | $65.3$ | $64.8$ | 38.8 | 36.3 |

**Q2**: Is augmenting synthetic data or high-return synthetic data critical for Offline policy?

**Q3**: Does our policy-decoupled data augmentation algorithm HIPODE outperform the conventional policy-dependent data augmentation methods?

**Q4**: In HIPODE, what roles do the negative sampling and transition selector components play?

In the following, we answer *Q1* in Section 5.1, showing the effectiveness and superiority of HIPODE by combining it with offline RL algorithms on MuJoCo (Todorov et al., 2012) tasks. Then, we present an ablation study in details in Section 5.2 to answer *Q2*. We answer *Q3* in Section 5.3 by comparing HIPODE with policy-dependent data augmentation methods. Finally, we answer *Q4* in Appendix C.6.

## 5.1 Performance on MuJoCo

**Evaluation settings**. We demonstrate the benefits of HIPODE on D4RL MuJoCo-v0 tasks (Fu et al., 2020), comparing with several baselines that generate augmented data for policy training, including CABI (Lyu et al., 2022), MOPO (Yu et al., 2020) and COMBO (Yu et al., 2021). Experimental details of baselines can be found in Appendix B.2.

**Main results**. Table 2 shows the results on 15 MuJoCo tasks comparing between the above-mentioned algorithms. HIPODE achieves remarkable improvements over baselines (TD3BC, CQL), and also significant gain outperforming SOTA data augmentation method (CABI), which confirm the effectiveness of HIPODE in handling these offline tasks. In this experiment, in order to make a fair comparison, we reproduce the CABI algorithm based on the same downstream policy learning process and the same hyper-parameters to demonstrate HIPODE's superiority. Under the premise of controlling downstream implementation and consistent hyper-parameters, the advantage of HIPODE performance all comes from the data augmentation process. On the other hand, compared to the reported results in (Lyu et al., 2022), our advantage remains consistent, as detailed in Appendix C.4. To show that HIPODE is extremely general, we significantly enhance IQL on Mujoco -v0 and Mujoco -v2 tasks and beat CABI as detailed in Appendix C.1 and Appendix C.2. HIPODE is also effective in the Adroit tasks and its details are presented in Appendix C.3.

HIPODE and policy-dependent methods, including previous model-based Offline RL methods have different objectives: while policy-dependent methods aim to enhance specific policies, HIPODE

focuses on strengthening various offline RL algorithms with minimal cost, which may sacrifice the enhancement effect on a specific policy. However, in order to gain a more comprehensive understanding of HIPODE's performance, we sill compare it to the model-based Offline RL methods. HIPODE's predominance over the model-based policy-dependent baseline algorithms (COMBO, MOPO) demonstrates its strength, as shown in Table 2. Noting that COMBO and MOPO need to access the true terminal function to ensure algorithm performance, whereas HIPODE achieves better performance without the need for such a function, by uniformly setting terminal flag of HIPODE's synthetic data to False. Since most recent Offline RL methods is evaluated on Mujoco -v2 tasks, we also combine HIPODE with IQL on 9 Mujoco -v2 tasks to compare the performance to the recent SOTA RAMBO-RL (Rigter et al., 2022). We find that HIPODE can enhance IQL on Mujoco -v2 tasks and has competitive total performance to RAMBO-RL. The details of experiments on Mujoco -v2 tasks can be find in Table 8 of Appendix C.2.

To show HIPODE indeed generates high-return transitions, we visualize the distribution of estimated discounted cumulative rewards of trajectories in the original dataset and synthetic data generated by HIPODE and CABI. Specifically, we train online SAC to converge to obtain the optimal value function $V^*$ as authoritative value function. For each state-action pair $(s, a)$, we use $r + V^*(s')$ to represent the discounted cumulative reward, where $r, s' \sim p(r, s'|s, a)$ are the true reward and next state in the environment respectively. Fig.4 illustrates the density of synthetic data generated by CABI and HIPODE on halfcheetah-medium-replay-v0, as well as the original dataset, where X axis represents $V^*$ and Y represents the density of synthetic transitions on $V^*$. Form the figure, the green shadow almost coincide with the red one, showing that CABI's data distribution almost coincide

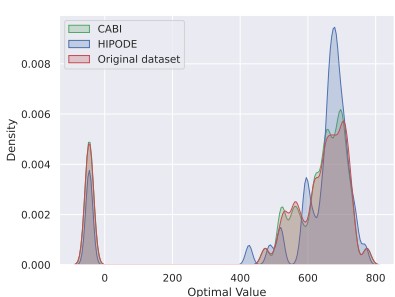

Figure 4: Density of different synthetic data and the original dataset.

with the original dataset, while HIPODE indeed generates more high-return data. In conjunction with the results in Table 2, the advantage of HIPIDE performance comes from more high-return data in augmentation process, which sequentially demonstrates that high-return data is more suitable rather than high diversity data as augmented data for Offline RL.

## 5.2 ABLATION STUDY

In this section, we aim to further investigate how the generated data improves downstream offline policy. We conduct ablation experiments from three perspectives: not generating synthetic data (Repeat), generating vanilla synthetic data (NoV), and generating high-return synthetic data (HIPODE). and the results are shown in Table 3.

Specifically, the difference of Repeat and HIPODE is that the synthetic data is replaced by 10% high-return data from the original dataset in Repeat. The difference between NoV and HIPODE is that the value maximization mechanism is removed in NoV, i.e., the return of generated data is not controlled. Generating high-return synthetic data is exactly HIPODE. Normalized score of enhancing TD3BC with these perspectives is shown as Repeat+TD3BC, NoV+ TD3BC, HIPODE+TD3BC in Table 3.

Table 3: Normalized average score and standard deviatio of generating different types of augmented data over 3 seeds on MuJoCo -v0 tasks.

| Task Name | Repeat +TD3BC | NoV +TD3BC | HIPODE +TD3BC | TD3BC |
|---|---|---|---|---|
| halfcheetah-m-e | 97.4±3.6 | 99.1±4.2 | 102.6±2.2 | 98.0±3.7 |
| hopper-m-e | 111.9±0.5 | 110.5±3.5 | 112.4±0.3 | 112±0.1 |
| walker2d-m-e | 38.0±64.4 | 102.5±5.6 | 105.3±4.0 | 105.4±3.9 |
| halfcheetah-m-r | 42.5±1.6 | 44.0±0.2 | 44.0±0.7 | 43.3±0.6 |
| hopper-m-r | 36.5±4.2 | 36.2±8.0 | 36.2±0.8 | 32.7±0.9 |
| walker2d-m-r | 18.0±14.4 | 30.0±4.1 | 36.4±11.1 | 19.6±8.6 |
| halfcheetah-m | 43.6±0.8 | 43.1±0.5 | 43.7±0.5 | 43.7±0.4 |
| hopper-m | 99.8±0.1 | 99.7±0.3 | 99.9±0.3 | 99.9±0.5 |
| walker2d-m | 79.5±2.7 | 79.7±3.1 | 80.1±0.7 | 79.7±1.8 |
| halfcheetah-r | 11.7±0.3 | 13.1±1.4 | 15.5±0.9 | 12.8±1.9 |
| hopper-r | 11.1±0.0 | 10.9±0.1 | 10.9±0.2 | 10.9±0.2 |
| walker2d-r | 1.9±0.4 | 2.1±0.7 | 6.6±1.0 | 0.4±0.1 |
| halfcheetah-e | 104.2±3.3 | 105.3±0.7 | 106.5±0.5 | 107.0±0.9 |
| hopper-e | 112.5±0.5 | 112.3±0.2 | 112.3±0.5 | 107.3±8.7 |
| walker2d-e | 78.1±48.5 | 104.9±5.8 | 106.6±5.2 | 98.7±6.0 |
| **Avg** | 59.1±9.7 | 66.2±2.6 | **67.9**±1.9 | 64.7±2.6 |

As the results in Table 3, Repeat+TD3BC, i.e., repeating high-return data in the dataset, brings little performance gain and even hurts performance on walker2d-medium-expert and walker2d-expert. Thus it's not effective for improving Offline

Table 4: Normalized score comparison of policy-dependent methods for data augmentation v.s. HIPODE and the baseline on MuJoCo -v0 tasks. We report average normalized score over 3 random seeds each task. Full results consists of -random and -expert tasks are presented in Appendix C.5.

| Task Name | MB+2.5 TD3BC | MB+0.001 TD3BC | MBPO | HIPODE+ TD3BC | TD3BC | BooT +CQL | CQL |
|---|---|---|---|---|---|---|---|
| halfcheetah-m-e | 26.0 | 73.0 | 9.7 | 102.6 | 98.0 | 5.0 | 94.8 |
| hopper-m-e | 1.1 | 42.6 | 56 | 112.4 | 112.0 | 0.8 | 111.9 |
| walker2d-m-e | 42.9 | 8.5 | 7.6 | 105.3 | 105.4 | 26.4 | 70.3 |
| halfcheetah-m-r | 45.8 | 23.1 | 47.3 | 44 | 43.3 | 4.3 | 42.5 |
| hopper-m-r | 4.8 | 20.9 | 49.8 | 36.8 | 32.7 | 5.0 | 28.2 |
| walker2d-m-r | 0.0 | 7.5 | 22.2 | 36.4 | 19.6 | 5.8 | 5.1 |
| halfcheetah-m | 45.4 | 36.7 | 28.3 | 43.7 | 43.7 | 30.0 | 39.2 |
| hopper-m | 0.7 | 30.2 | 4.9 | 99.9 | 99.9 | 79.8 | 30.3 |
| walker2d-m | 4.3 | 16.9 | 12.7 | 80.1 | 79.7 | 6.4 | 66.9 |
| **Total** | 171.0 | 259.4 | 238.5 | **661.2** | 634.3 | 163.5 | 489.2 |

RL performance. Besides, NoV+TD3BC achieves an improvement over TD3BC, indicating the importance of generating new synthetic data for data augmentation. However, the performance of NoV+TD3BC is worse than HIPODE, indicating the importance of generating high-return data. To summarize, the result suggests that generating synthetic data is more effective than simply repeat data, but the pursuit of generating higher return synthetic data can bring more significant performance improvements for downstream Offline RL performance.

## 5.3 COMPARISON WITH POLICY-DEPENDENT DATA AUGMENTATION METHODS

In policy-dependent data augmentation methods, the data generation process is tightly tied to the downstream Offline RL policy, which limits the applicability of the generated data. In this section we aim to illustrate the strength of our policy-decoupled data augmentation method, compared to policy-dependent methods on different downstream Offline RL policies. Specifically, on the downstream TD3BC algorithm, we evaluate the effect of data generated with some model-based policy dependent algorithms; on the downstream CQL algorithm, we analyze the effect of the more advanced policy dependent algorithm Boot (Wang et al., 2022).

We first evaluate the performance of dynamics-model-enhanced TD3BC based on the data generated by a previously trained dynamics model, by rolling-out current TD3BC policy on the dynamics model. The results are reported as MB+$\alpha$TD3BC in Table 4, where $\alpha$ is a hyper-parameter in TD3BC (Fujimoto & Gu, 2021). We also compare offline MBPO with HIPODE since it can also be seen as a method directly using dynamics-model-generated data as augmented data. The difference between model based TD3BC and MBPO is that TD3BC has a behaviour cloning restrict on it's critic (Fujimoto & Gu, 2021) while MBPO (Janner et al., 2019) dose not. Results in Table 4 indicate that using dynamics-model-generated data as augmentation will damage the offline agent, and such damage can be mitigated when the policy of the offline agent is closed to the behavioural policy of the dataset. This suggests that the damage is caused by the difference between the policy the dynamics model trained on and the policy it generates data on, which these model-based policy-dependent methods fail to address.

We then directly take results report in (Wang et al., 2022) to form the BooT+CQL column in Table 4. BooT+CQL means directly using synthetic data generated by BooT on CQL. The results show that synthetic data generated by Boot has poor results as augmented data combined with CQL. This indicates that synthetic data generated by a policy-dependent data augmentation method can damage another offline agent. In contrast, HIPODE is policy-decoupled and our augmented data can benefit different offline agent without changing, as shown in Table 2.

In summary, synthetic data generated by policy-dependent data augmentation methods may have a detrimental effect on Offline RL processes, while the synthetic data generated by HIPODE can improve their performance, demonstrating the superiority of HIPODE.

## 6 CONCLUSIONS AND LIMITATIONS

In this paper, we investigate the issues of data augmentation for Offline RL. We conduct extensive experiments to demonstrate that, in the context of Offline RL, high-return data is a more suitable choice for augmented data than high-diversity data when the authority of the data is the same. Based on this observation, we propose a novel data augmentation method called HIPODE, which selects states with higher values as augmented data. This ensures that the synthetic data is both authentic and of high-return and is generated in a policy-decoupled manner. Our experimental results on D4RL benchmarks demonstrate that HIPODE significantly improves the performance of several widely used model-free Offline RL baselines without changing the augmented data, thereby achieving policy-decoupled data augmentation and demonstrating superiority over policy-dependent methods. Furthermore, HIPODE outperforms SOTA policy-decoupled data augmentation methods for Offline RL, demonstrating the benefits by generating high-return data.

However, HIPODE is outperformed by vanilla model-based Offline RL methods (e.g., MBPO) on -random datasets because the value penalty is excessively strict on those datasets. We believe that adjusting the penalty weight to be state-dependent instead of initially setting it to a fixed value is a potential solution to this issue, which we leave for future work.

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
