## A    PSEUDOCODE OF HIPODE

In this section, we summarize HIPODE in Algorithm 1, where $N(\mathcal{D}_{\text{env}})$ is the amount of data in the original dataset. Line 5-7 refers to selecting the next state with negative sampling, described in Section 4.1 for generating synthetic next states that are close to the dataset distribution and have the potential for the highest value. Line 8-10 and 12 refer to replenishing transitions with an action selector, as described in Section 4.2 for generating synthetic actions and rewards. Line 12-13 refers to merging the synthetic data with the original dataset for downstream offline policy training to obtain the offline policy $\pi$.

---

**Algorithm 1:** HIPODE

---

1  **Input:** Offline dataset $\mathcal{D}_{\text{env}} = \{(s, a, r, s')\}$, penalty wieght $\alpha$, synthetic rate $\eta$, action selecting rate $\lambda$, number of candidate next states $n$

2  **Output:** Policy $\pi$

3  Train state transition model $D_\psi(s, z)$ by minimizing Eq.1, value estimator $V_\theta(s)$ by minimizing Eq.4, dynamics model $\hat{p}_w(s'|s, a)$ by minimizing Eq.6 and inverse action model $D_\epsilon(s, s', z)$ as well as inverse reward model $D_\zeta(s, s', z)$ by minimizing Eq.5 and a similar loss for reward generation respectively

4  **repeat**

5  $\quad$ Sample a batch of $s$ from $\mathcal{D}_{\text{env}}$

6  $\quad$ Sample $\tilde{S}'_{\text{cand}}$ containing $n$ $\tilde{s}'_i$ from $\tilde{s}'_i \sim D_\psi(s, z), z \sim \mathcal{N}(0, \mathrm{I})$

7  $\quad$ $\tilde{s}' = \arg\max_{\tilde{s}'_{\text{cand}}} V(\tilde{s}'_{\text{cand}}), \tilde{s}'_{\text{cand}} \in \tilde{S}'_{\text{cand}}$

8  $\quad$ Sample actions $\tilde{a} \sim D_\epsilon(s, s', z), z \sim \mathcal{N}(0, \mathrm{I})$ and sample rewards $\tilde{r} \sim D_\zeta(s, s', z), z \sim \mathcal{N}(0, \mathrm{I})$

9  $\quad$ Sample $\tilde{s}'_{\text{dyna}}$ from $\tilde{s}'_{\text{dyna}} \sim \tilde{p}_w(\tilde{s}'_{\text{dyna}}|s, a)$

10  **until** *reaching maximum generating amount, which is* $\eta N(\mathcal{D}_{\text{env}})$;

11  Select top $\lambda$-portion authentic actions to construct $D_{\text{synthetic}}$ with least $\|\tilde{s}'_{\text{dyna}} - \tilde{s}'\|$

12  Merge the synthetic dataset and the original dataset $D = D \cup D_{\text{synthetic}}$

13  Use any model-free offline policy learning algorithm to obtain $\pi$

14  **return** $\pi$

---

## B    DETAILED SETTINGS OF EXPERIMENTS

### B.1    D4RL TASKS

In this section, we describe details about the MuJoCo and Adroit tasks in the D4RL (Fu et al., 2020) benchmark suite, on which we evaluate HIPODE.

**MuJoCo** contains a series of continuous locomotion tasks. Among them, Walker2d is a bipedal robot control task, where the goal is to maintain the balance of robot body and move as fast as possible. Hopper is a single-legged robot control task where the goal is to make the robot jump as fast as possible. Halfcheetah is to make a simulated robot perform a running motion that resembles a cheetah's movement, while trying to maximize the distance traveled within a fixed time period. In the MuJoCo-v0 tasks of D4RL, each environment has 5 different types of datasets: expert, medium-expert, medium-replay, medium and random. **Expert**: a large amount of data collected by a well-trained SAC agent. **Medium**: a large amount of data collected by a early-stopped SAC agent. **Medium-expert**: a large amount of mixed data of medium and expert at a 50-50 ratio. **Medium-replay**: replay buffer of a early-stopped SAC agent. **Random**: a large amount of data collected by a random policy.

**Adroit** contains a series of continuous and sparse-reward robotic environment to control a 24-DoF simulated Shadow Hand robot to twirl a pen, hammer a nail, open a door or grab a ball. These environments are even hard for online learning due to sparse rewards and exploration challenges (Fu et al., 2020). There are three types of datasets for each environment: expert, human and cloned. Among them, we evaluate HIPODE on human datasets, where a small number of demonstrations operated by a human (25 trajectories per task) is collected.

We report normalized score based on the protocol described in (Fu et al., 2020). A score of 0 represents the average return of random policies, and a score of 100 represents the return of a domain-specific expert. In our experiments, all score is the final performance of the downstream

offline reinforcement learning (Offline RL) algorithms, which is the average cumulative reward of ten final policy rollouts.

## B.2 EXPERIMENTAL DETAILS

In this setcion we provied experiment details for the experiments in Section 3 and Section 5.1. **Takeaway experimental details.** We use the CVAE to learn the behavioral policy and use CQL as the high-return-augmentation-policy for halfcheetah-random-v0. Additionally, TD3BC was utilized as the high-return-augmentation-policy for halfcheetah-medium-replay-v0. Our principle for selecting the return augmentation policy was to ensure that the testing performance of the augmentation policy slightly surpass the highest performance in the dataset. The ratio of augmented data (true data from the augmentation policy) to original data is 1:2.

**Main experiments' details.** We demonstrate HIPODE on D4RL MuJoCo-v0 tasks (Fu et al., 2020), comparing with several baselines that generate augmented data for policy training:

- **CABI** (Lyu et al., 2022), the SOTA policy-decoupled data augmentation algorithm for Offline RL. we reproduce CABI following their paper (Lyu et al., 2022) based on CORL (Tarasov et al., 2022).

- **COMBO** (Yu et al., 2021) and **MOPO** (Yu et al., 2020), two widely studied model-based Offline RL methods, which are policy-dependent. we re-run COMBO (Yu et al., 2021) using code of (Sun, 2023), and take the reported results of MOPO directly from the original paper (Yu et al., 2020). Additionally, We re-run MOPO on expert datasets using OfflineRL-Kit (Sun, 2023).

To ensure the fairness of the comparison, we implement both HIPODE and the downstream Offline RL algorithms (TD3BC and CQL) based on CORL (Tarasov et al., 2022). For the implementation of HIPODE, we tune the synthetic rate $\eta$, which represents the proportion of synthetic in each offline policy learning batch, as a main hyper-parameter. Most of the tasks have a similar score from a synthetic rate $< 0.5$. Besides, we choose action selecting rate $\lambda$ from $\{0.2, 1.0\}$, and set the penalty weight $\alpha$ 1.0 as default. We present detailed discussion about benchmark tasks and implementation in Appendix B.1 and Appendix B.4.

Table 5: Hyper-parameters of HIPODE+CQL and HIPODE+TD3BC.

| HIPODE+CQL Task Name | synthetic rate $\eta$ | selecting rate $\lambda$ | candidate number $n$ | penalty weight $\alpha$ | penalty scope $\sigma$ | CQL $\alpha$ |
|---|---|---|---|---|---|---|
| halfcheetah-e | 0.2 | 0.2 | 10 | 1.0 | 1.0 | 10.0 |
| halfcheetah-m-e | 0.1 | 0.2 | 10 | 1.0 | 1.0 | 10.0 |
| halfcheetah-m-r | 0.2 | 1.0 | 10 | 1.0 | 1.0 | 10.0 |
| halfcheetah-m | 0.2 | 1.0 | 10 | 1.0 | 1.0 | 10.0 |
| halfcheetah-r | 0.1 | 0.2 | 10 | 1.0 | 1.0 | 10.0 |
| hopper-e | 0.2 | 1.0 | 10 | 1.0 | 1.0 | 10.0 |
| hopper-m-e | 0.2 | 0.2 | 10 | 1.0 | 1.0 | 10.0 |
| hopper-m-r | 0.1 | 0.2 | 10 | 1.0 | 1.0 | 10.0 |
| hopper-m | 0.2 | 0.2 | 10 | 1.0 | 1.0 | 10.0 |
| hopper-r | 0.2 | 1.0 | 10 | 1.0 | 1.0 | 10.0 |
| walker2d-e | 0.2 | 0.2 | 10 | 1.0 | 1.0 | 10.0 |
| walker2d-m-e | 0.05 | 0.2 | 10 | 1.0 | 1.0 | 10.0 |
| walker2d-m-r | 0.4 | 1.0 | 10 | 1.0 | 1.0 | 10.0 |
| walker2d-m | 0.2 | 1.0 | 10 | 1.0 | 1.0 | 10.0 |
| walker2d-r | 0.2 | 1.0 | 10 | 1.0 | 1.0 | 10.0 |

## B.3 HYPER-PARAMETERS

In this section, we provide the key hyper-parameters used for HIPODE in the main experiments. They are listed in Table 5 and Table 6. Unless specified otherwise, all results in this paper are obtained using the hyper-parameters listed here. To reproduce CABI, we follow the hyper-parameters provided in (Lyu et al., 2022). For COMBO, we use the Offline-RL-Kit's code (Sun, 2023) and follow (Yu et al., 2021) to set the real ratio to 0.8 for both walker2d-medium-expert and walker2d-expert while

Table 6: Hyper-parameters of HIPODE+TD3BC on 15 MuJoCo -v0 tasks and 4 Adroit-human-v0 tasks.

| Task Name | synthetic rate $\eta$ | selecting rate $\lambda$ | candidate number $n$ | penalty weight $\alpha$ | penalty scope $\sigma$ | TD3BC $\alpha$ |
|---|---|---|---|---|---|---|
| halfcheetah-e | 0.15 | 0.2 | 10 | 1.0 | 1.0 | 2.5 |
| halfcheetah-m-e | 0.15 | 0.2 | 10 | 1.0 | 1.0 | 2.5 |
| halfcheetah-m-r | 0.5 | 0.2 | 10 | 1.0 | 1.0 | 2.5 |
| halfcheetah-m | 0.2 | 0.2 | 10 | 1.0 | 1.0 | 2.5 |
| halfcheetah-r | 0.7 | 0.2 | 10 | 1.0 | 1.0 | 2.5 |
| hopper-e | 0.2 | 0.2 | 10 | 1.0 | 1.0 | 2.5 |
| hopper-m-e | 0.5 | 0.2 | 10 | 1.0 | 1.0 | 2.5 |
| hopper-m-r | 0.5 | 0.2 | 10 | 1.0 | 1.0 | 2.5 |
| hopper-m | 0.1 | 0.2 | 10 | 1.0 | 1.0 | 2.5 |
| hopper-r | 0.15 | 0.2 | 10 | 1.0 | 1.0 | 2.5 |
| walker2d-e | 0.2 | 0.2 | 10 | 1.0 | 1.0 | 2.5 |
| walker2d-m-e | 0.05 | 0.2 | 10 | 1.0 | 1.0 | 2.5 |
| walker2d-m-r | 0.5 | 0.2 | 10 | 1.0 | 1.0 | 2.5 |
| walker2d-m | 0.15 | 0.2 | 10 | 1.0 | 1.0 | 2.5 |
| walker2d-r | 0.4 | 0.2 | 10 | 1.0 | 1.0 | 2.5 |
| door-human | 0.4 | 0.2 | 10 | 1.0 | 1.0 | 0.001 |
| hammer-human | 0.2 | 0.2 | 10 | 1.0 | 1.0 | 0.001 |
| pen-human | 0.8 | 0.2 | 10 | 1.0 | 1.0 | 0.001 |
| relocate-human | 0.4 | 0.2 | 10 | 1.0 | 1.0 | 0.001 |

0.5 for other tasks. For all other hyper-parameters in COMBO and MOPO -expert, we use the default hyper-parameters provided in the Offline-RL-Kit's code [1].

### B.4 IMPLEMENTATION DETAILS

In this section, we describe details of the implementation to our experiments.

**Data compression.** For data augmentation, generating and storing large amount of synthetic data into a static dataset before the downstream Offline RL process can be resource-intensive. To address this issue, in practice, HIPODE is integrated into the downstream Offline RL process by generating synthetic data in real-time during the Offline RL process. This technique compress the size of synthetic data to the parameters of several generative models in HIPODE. In the data generating process, the synthetic rate $\eta$ denotes the rate of synthetic data in every batch fed into downstream Offline RL algorithms. Therefor, a batch of size $N$ contains $\eta N$ synthetic transitions and $(1 - \eta)N$ real transitions.

**Models in HIPODE.** Here we describe the details about the models in HIPODE. There are 5 independent models in HIPODE:

- **Value network.** The value network is implemented using a Multi-Layer Perceptron (MLP) with one hidden layer of 256 units. We update the target value network every 2 gradient steps using the soft update method $\bar{\theta} = \tau\theta + (1 - \tau)\bar{\theta}$, where $\tau = 0.005$ is the update rate.

- **Inverse action model.** The inverse action model is implemented using a CVAE (Sohn et al., 2015) to generate an action from a given state and next state. The encoder and decoder of the CVAE both have one hidden layer of 750 units, and the latent dimension is twice the state dimension of each task.

- **Inverse reward model.** The inverse reward model is implemented using a CVAE to generate a reward from a given state and next state. Its structure is similar to the inverse action model, but the two models are trained separately. Together, they are referred to as the 'inverse dynamics model'.

[1]CORL code URL: `https://github.com/tinkoff-ai/CORL`

- **Forward dynamics model.** The forward dynamics model predicts the next state from a given current state and action. For a fair comparison, the implementation of the forward dynamics model is identical for both CABI and HIPODE and we refer to the implementation of the dynamics model in the D3RLPY (Seno & Imai, 2022) library [2] for the implementation of this part.
- **State transition model.** The state transition model is implemented using a CVAE to generate the next state from a given current state, following (Zhang et al., 2022). Its structure is the same as that of the inverse action model, except that the input dimension of the encoder and the output dimension of the decoder are different.

From the details of the HIPODE models, it can be seen that, although the data generating process is integrated into the downstream Offline RL process, the training process and data generating process of all the models in HIPODE is decoupled from the downstream Offline RL process, thus achieving policy-decoupled data augmentation.

## C  ADDITIONAL RESULTS

### C.1  HIPODE+IQL ON MUJOCO -V0 TASKS

Following the experimental setup described in Section 5, we conduct experiments on 9 Mujoco -v0 tasks based on the IQL algorithm. The results are in Table 7. Since we combine HIPODE with the CORL IQL code, we also provide the reproduced results of the CORL IQL ('IQL(CORL)' in the table) to ensure experimental fairness and provide reference. The results demonstrate that HIPODE can significantly enhance IQL on Mujoco -v0 tasks.

To gain a deeper understanding of HIPODE's performance, we also compare it with a latest method, SPOT (Wu et al., 2022). We run the CORL SPOT code with the hyper-parameters provided in its paper.

Table 7: Average normalized score and standard deviation of HIPODE+IQL v.s. baseline methods over 3 random seeds on Mujoco-v0 tasks. In the table, CABI+IQL and IQL(paper) are directly taken from the CABI paper (Lyu et al., 2022), IQL(CORL) represents score we produced using the CORL code, $\eta$ and $\lambda$ represent the synthetic rate and selecting rate of HIPODE+IQL on Mujoco -v0 tasks respectively.

| | HIPODE +IQL | CABI +IQL | IQL (paper) | IQL (CORL) | SPOT | $\eta$ | $\lambda$ |
|---|---|---|---|---|---|---|---|
| halfcheetah-m-e | 88.7±9.7 | **96.7**±1.3 | 89.0±0.7 | 86.6±5.1 | 75.7±11.1 | 0.05 | 0.2 |
| hopper-m-e | **112.8**±0.1 | **112.8**±0.2 | 111.5±1.0 | 110.8±2.6 | **112.6**±0.4 | 0.2 | 0.2 |
| walker2d-m-e | **105.5**±2.4 | 104.8±1.0 | 99.7±2.9 | 97.6±9.8 | 88.3±16.4 | 0.1 | 0.2 |
| halfcheetah-m | **42.7**±0.5 | 41.6±0.1 | 41.2±0.1 | 42.3±0.6 | 40.7±0.2 | 0.2 | 0.2 |
| hopper-m | **94.5**±7.0 | 40.0±12.9 | 30.7±0.0 | 64.0±31.9 | 42.6±21.4 | 0.05 | 0.2 |
| walker2d-m | 67.1±2.9 | 55.1±2.3 | 50.8±7.7 | 64.8±8.1 | **75.3**±6.7 | 0.1 | 1.0 |
| halfcheetah-m-r | **42.5**±0.2 | 42.2±0.2 | 40.5±0.4 | 41.2±1.1 | 40.7±1.5 | 0.2 | 0.2 |
| hopper-m-r | 35.2±0.4 | **36.8**±0.4 | 33.4±1.1 | 31.4±2.8 | 27.4±1.5 | 0.05 | 1.0 |
| walker2d-m-r | **22.1**±0.9 | 17.2±0.8 | 15.8±1.7 | 14.1±2.8 | 21.3±8.2 | 0.2 | 0.2 |
| **Total** | **611.1** | 547.2 | 512.6 | 552.8 | 524.6 | - | - |

### C.2  HIPODE ON MUJOCO -V2 TASKS

Since most of the recent Offline RL works are evaluated on Mujoco -v2 tasks, we also conduct experiments on Mujoco -v2 tasks to further further test the generalization of HIPODE. The results summarized in Table 8 demonstrate that HIPODE can also enhance IQL on Mujoco -v2 tasks. To our surprise, HIPODE+IQL achieve competitive total score compared with RAMBO-RL, the state-of-the-art model-based Offline RL method.

---

[2]D3RLPY code URL: `https://github.com/takuseno/d3rlpy`

Table 8: Average normalized score and standard deviation of HIPODE+IQL v.s. baseline methods over 4 random seeds on Mujoco-v2 tasks. In the table, IQL(paper) are results directly taken from IQL paper (Kostrikov et al., 2021), IQL(CORL) represents the score we produced using the CORL code, $\eta$ and $\lambda$ represent the synthetic rate and selecting rate of HIPODE+IQL on Mujoco -v2 tasks respectively and RAMBO-RL's results is taken directly from its paper (Rigter et al., 2022).

|  | HIPODE+IQL | IQL(paper) | IQL(CORL) | $\eta$ | $\lambda$ | RAMBO-RL |
|---|---|---|---|---|---|---|
| halfcheetah-m-e | 91.8±4.9 | 86.7 | 91.8 | 0.05 | 0.2 | **93.7** |
| hopper-m-e | **108.2**±5.0 | 91.5 | 104 | 0.2 | 1.0 | 83.3 |
| walker2d-m-e | 111.7±1.0 | 109.6 | **112.1** | 0.05 | 0.2 | 68.3 |
| halfcheetah-m | 49.1±0.3 | 47.4 | 48.3 | 0.2 | 0.2 | **77.6** |
| hopper-m | 66.6±6.9 | 66.3 | 64.9 | 0.1 | 0.2 | **92.8** |
| walker2d-m | 83.6±3.5 | 78.3 | 80.1 | 0.1 | 0.2 | **86.9** |
| halfcheetah-m-r | 44.9±0.2 | 44.2 | 43.9 | 0.1 | 1 | **68.9** |
| hopper-m-r | **101.1**±1.0 | 94.7 | 97.0 | 0.05 | 1 | 96.6 |
| walker2d-m-r | 79.6±3.5 | 73.9 | 65.8 | 0.1 | 0.2 | **85.0** |
| tot | 736.6 | 692.6 | 707.9 | - | - | **753.1** |

Table 9: Average normalized score and standard deviation of HIPODE+TD3BC v.s. TD3BC over 3 random seeds on Adroit-human tasks.

| Task Name | HIPODE+TD3BC | TD3BC |
|---|---|---|
| door-human | $2.7 \pm 2.6$ | $1.3 \pm 1.2$ |
| hammer-human | $3.5 \pm 4.0$ | $3.9 \pm 5.6$ |
| pen-human | $85.0 \pm 14.3$ | $60.8 \pm 14.0$ |
| relocate-human | $0.2 \pm 0.1$ | $0.1 \pm 0.1$ |
| **Total** | **91.4** | 66.1 |

## C.3 HIPODE ON ADROIT TASKS

To further demonstrate the effectiveness of HIPODE, we also conduct experiments on Adroit-human tasks to evaluate its performance in a more challenging setting with limited human demonstrations and sparse rewards. As shown in Table 9, HIPODE is effective, improving the baseline TD3BC on three out of four challenging Adroit tasks. However, the performance of HIPODE+TD3BC on Adroit-human datasets is less effective. We attribute this to the poor performance of the baseline offline agent on Adroit tasks due to the tasks' complexity (Fu et al., 2020).

## C.4 HIPODE V.S. REPORTED CABI RESULTS

In Table 2, we report score of reproduced CABI for a fair comparison. However, our reproduced results are slightly worse than those reported in the original CABI paper. To provide a more comprehensive view of performance, we list the results reported in the original CABI paper in Table 10. As shown in Table 10, HIPODE's advantage compared to CABI is still significant when combined with CQL, while comparable when combined with TD3BC.

Additionally, to further demonstrate the strong potential of HIPODE, we also conduct experiments tuning the penalty weight $\alpha$. We find that adjusting the penalty weight $\alpha$ on some tasks (e.g. walker2d-expert-v0 in Table 13) can improve the performance, leading to a higher total score than that of reported CABI+TD3BC. For the sake of fair comparison and ease of use, we only report the results of penalty-weight-non-tuned experiments in this paper.

## C.5 FULL RESULTS FOR POLICY-DEPENDENT METHODS V.S. HIPODE

In this section, we present the results of all 15 MuJoCo tasks to compare the performance of HIPODE with policy-dependent data augmentation methods, and the results are shown in Table 11. We begin by evaluating the performance of dynamics-model-enhanced TD3BC, using data generated by a previously trained dynamics model. Specifically, we roll out the current TD3BC policy on the

Table 10: Average normalized score of HIPODE over 3 random seeds v.s. reported CABI on 15 MuJoCo -v0 tasks.

| Task Name | HIPODE +CQL | CABI +CQL | HIPODE +TD3BC | CABI +TD3BC |
|---|---|---|---|---|
| halfcheetah-m-e | 101.8 | 35.3 | 102.6 | 105.0 |
| hopper-m-e | 112.1 | 112.0 | 112.4 | 112.7 |
| walker2d-m-e | 91.7 | 107.5 | 105.3 | 108.4 |
| halfcheetah-m-r | 44.8 | 44.6 | 44.0 | 44.4 |
| hopper-m-r | 33.5 | 34.8 | 36.2 | 31.3 |
| walker2d-m-r | 21.7 | 21.4 | 36.4 | 29.4 |
| halfcheetah-m | 39.3 | 42.4 | 43.7 | 45.1 |
| hopper-m | 30.4 | 57.3 | 99.9 | 100.4 |
| walker2d-m | 79.9 | 62.7 | 80.1 | 82.0 |
| halfcheetah-r | 22.9 | 30.2 | 15.5 | 15.1 |
| hopper-r | 10.4 | 10.7 | 10.9 | 11.9 |
| walker2d-r | 16.5 | 7.3 | 6.6 | 6.4 |
| halfcheetah-e | 108.5 | 99.2 | 106.5 | 107.6 |
| hopper-e | 112.2 | 112.0 | 112.3 | 112.4 |
| walker2d-e | 109.9 | 110.2 | 106.6 | 108.6 |
| **Total** | **935.6** | **887.6** | **1019.0** | **1020.7** |

dynamics model. The results are reported in Table 11 as MB+$\alpha$TD3BC, where $\alpha$ is a hyper-parameter in TD3BC (Fujimoto & Gu, 2021). The MB+$\alpha$TD3BC results, together with those from MBPO, suggest that using dynamics-model-generated data as augmentation can harm the offline agent. However, we find that the damage can be reduced when the policy of the offline agent is close to the behavioural policy of the dataset. This suggests that the damage is caused by the mismatch between the policy that the dynamics model is trained on and the policy it uses to generate data, which cannot be addressed by model-based policy-dependent methods.

We then conduct experiments on a more advanced policy-dependent data augmentation method, BooT (Wang et al., 2022). We directly take results report in (Wang et al., 2022) to form the BooT+CQL column in Table 4. BooT+CQL presents the use of synthetic data generated by BooT as augmentation data for CQL. The results indicate that the synthetic data generated by BooT performs poorly when used as augmentation data combined with CQL.This suggests that synthetic data generated by a policy-dependent data augmentation method can have a detrimental effect on Offline RL algorithms. In contrast, HIPODE's synthetic data can benefit them without causing harm, as demonstrated in the HIPODE+TD3BC and HIPODE+CQL column in Table 11.

In summary, the synthetic data generated by policy-dependent data augmentation methods can have a negative impact on Offline RL processes, while HIPODE' synthetic data can improve them, demonstrating the superiority of HIPODE.

### C.6 More Ablation Study

In this section, we investigate how HIPODE enhances downstream offline policy learning performance by examining two key components: the negative sampling mechanism and the state transition model. We analyze the impact of these components as independent variables of downstream offline policy learning performance.

**Is the state transition model critical?** To investigate the importance of state transition model, we remove the state transition model and randomly generate candidate next states inside the hypercube formed by the states in the original dataset with the other mechanisms and hyper-parameters stay the same. Results in Table 12 shows that removing state transition model severely drops compare to the baseline. In terms of the results in Table 12, although the negative sampling mechanism penalizes the OOD states in the no-state-transition-model condition, randomly choosing candidate next states can damage the downstream offline policy learning process, demonstrating the necessity of a state transition model. We believe the following reasons are responsible for this:

Table 11: Full results of normalized score comparison of policy-dependent methods for data augmentation v.s. HIPODE and the baseline. We report average score over 3 random seeds each task.

| Task Name | mb+2.5 TD3BC | mb+0.001 TD3BC | MBPO | HIPODE+ TD3BC | TD3BC | BooT +CQL | CQL | HIPODE+ CQL |
|---|---|---|---|---|---|---|---|---|
| halfcheetah-m-e | 26.0 | 73.0 | 9.7 | 102.6 | 98.0 | 5.0 | 94.8 | 99.5 |
| hopper-m-e | 1.1 | 42.6 | 56 | 112.4 | 112.0 | 0.8 | 111.9 | 112.1 |
| walker2d-m-e | 42.9 | 8.5 | 7.6 | 105.3 | 105.4 | 26.4 | 70.3 | 94.0 |
| halfcheetah-m-r | 45.75 | 23.1 | 47.3 | 44 | 43.3 | 4.3 | 42.5 | 46.0 |
| hopper-m-r | 4.8 | 20.9 | 49.8 | 36.8 | 32.7 | 5.0 | 28.2 | 34.7 |
| walker2d-m-r | 0.0 | 7.5 | 22.2 | 36.4 | 19.6 | 5.8 | 5.1 | 21.7 |
| halfcheetah-m | 45.4 | 36.7 | 28.3 | 43.7 | 43.7 | 30.0 | 39.2 | 39.3 |
| hopper-m | 0.7 | 30.2 | 4.9 | 99.9 | 99.9 | 79.8 | 30.3 | 30.4 |
| walker2d-m | 4.3 | 16.9 | 12.7 | 80.1 | 79.7 | 6.4 | 66.9 | 75.7 |
| Total | 171.0 | 259.4 | 238.5 | **661.2** | 634.3 | 163.5 | 489.2 | 553.4 |
| halfcheetah-r | 27.2 | 2.3 | 30.7 | 15.5 | 12.8 | - | 17.0 | 23.1 |
| hopper-r | 4.7 | 9.6 | 4.5 | 10.9 | 10.9 | - | 10.4 | 10.4 |
| walker2d-r | 0.1 | 1.3 | 13.6 | 6.4 | 0.4 | - | 1.7 | 10.5 |
| Total | 203.0 | 272.6 | 287.3 | **694.0** | 658.4 | | 518.3 | 597.4 |
| halfcheetah-e | -2.1 | 106.8 | - | 106.5 | 107.0 | - | 109.8 | 109.1 |
| hopper-e | 1.3 | 111.9 | - | 112.3 | 107.3 | - | 112.0 | 112.2 |
| walker2d-e | 49.3 | 84.4 | - | 106.6 | 98.7 | - | 104.7 | 109.3 |
| Total | 251.0 | 575.7 | - | **1019.4** | 971.4 | - | 844.9 | 929.0 |

Table 13: Normalized score of TD3BC combined with different penalty weight for data augmentation. The numbers in the headline denotes the penalty weight. We report average score over 3 random seeds each task.

| Task Name | -1+TD3BC | 0+TD3BC | 1+TD3BC | 2+TD3BC | 4+TD3BC | 8+TD3BC | TD3BC |
|---|---|---|---|---|---|---|---|
| walker2d-e | 105.2±2.1 | 102.9±8.1 | 106.6±5.2 | 105.2±2.2 | 106.6±1.0 | 109.1±0.2 | 98.7 |
| halfcheetah-m-r | 43.0±0.3 | 40.6±2.2 | 44.0±0.4 | 44.0±0.7 | 43.8±0.6 | 43.8±0.3 | 43.3 |

(1) The value function is not authentic on randomly sampled next states, so the value is not very effective; (2) few in-support next states are generate so the winning next state may still be an OOD state resulting in a inauthentic synthetic transition; (3) The synthetic transition can lead the policy to a randomly state during evaluation. Hence, state transition model is significantly critical to ensure an authentic augmentation. This resembles the necessity of a behavioral policy in CABI and is consistent with their conclusions (Lyu et al., 2022).

Table 12: Normalized results over 3 random seeds of HIPODE with randomly sampling next state v.s. HIPODE with CVAE next state.

| Name | HIPODE(no stm) +TD3BC | HIPODE+ TD3BC | TD3BC |
|---|---|---|---|
| halfcheetah-m-r | 0.6±0.1 | 44 | 43.3 |
| halfcheetah-m | 27.3±2.9 | 43.7 | 43.7 |
| halfcheetah-m-e | 58.2±3.0 | 102.6 | 98.0 |

**Is negative sampling critical?** In negative sampling, the penalty weight controls the severity of the penalty added to the value of OOD states. A larger penalty weight makes it less likely for HIPODE to generate an OOD next state. We change the penalty weight in $\{-1, 0, 1, 2, 4, 8\}$ and run the downstream offline policy learning algorithm, without changing the other hyper-parameters on walker2d-expert-v0 and halfcheetah-medium-replay-v0. The results in Table 13 show that penalty weights greater than 0 outperforms the others, but overall, the score difference is marginal. This indicates that penalizing the value function on OOD states can indeed benefit downstream offline policy learning process. On the other hand, the state transition model generates candidate states near the dataset, which makes the effect of the penalty insignificant.

## C.7 AUGMENTED DATA ANALYSIS

Figure 5 shows the Synthetic data produced by HIPODE trained on different datasets. We present two metrics to measure the synthetic data, which are the L2 distance from the original data in the dataset and the dynamics mean square error (MSE) with respect to the ground-truth data. A larger L2 distance indicates that the generated data is more novel, and a smaller dynamics MSE signifies that the generated data is more authoritative. The results show that the data generated by HIPODE on halfcheetah-random-v0 and halfcheetah-medium-replay-v0 are relatively more novel while remaining authoritative. This explains the improvement from HIPODE on halfcheetah-random-v0 and halfcheetah-medium-replay-v0 from another perspective, disregarding the return of the data.

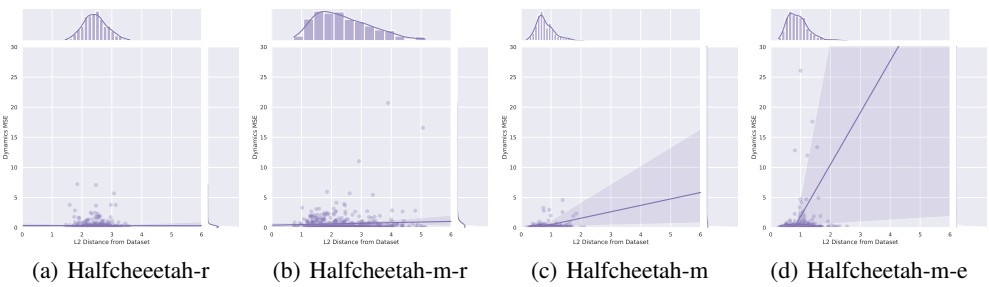

(a) Halfcheeetah-r      (b) Halfcheetah-m-r      (c) Halfcheetah-m      (d) Halfcheetah-m-e

Figure 5: L2 distance from training data and dynamics accuracy of HIPODE on 4 tasks of Halfcheetah. In the figure, -r denotes random-v0, -m-r denotes medium-replay-v0, -m denotes medium-v0 and -m-e denotes -medium-expert.