# OpenReview forum: "HIPODE: Enhancing Offline Reinforcement Learning with High-Quality Synthetic Data from a Policy-Decoupled Approach"
_ICLR.cc/2024/Conference — Submitted to ICLR 2024_

### Official Review · Reviewer_Pxud · 2023-10-22

**Soundness:** 2 fair
**Presentation:** 3 good
**Contribution:** 3 good
**Rating:** 6
**Confidence:** 4

**Summary:**

This paper proposes a method, HIPODE, for generating synthetic data that can improve the performance of offline RL algorithms. HIPODE generates high-return samples by generating multiple next-state candidates from a trained CVAE and selecting the sample with the highest value estimate. An L2 penalty term is added to the critic loss to prevent overestimating OOD state values. From the sampled high-return states, actions and rewards are computed using an inverse dynamics model. Finally, the authenticity of the actions is checked using a forward dynamic model, and unreliable actions are discarded.

**Strengths:**

### Originality

* The authors conducted experiments that show the importance of high-return trajectories.

* They devised a clever next-state generation method based on negative sampling that enables HIPODE to generate high-valued next-states without overestimating the values too much.

### Quality

* The paper presents multiple experimental results that can show the effectiveness of HIPODE.

* The Introduction and Related Work sections provide a nice overview of existing Offline RL algorithms.

### Clarity

The paper is overall well-written and is easy to understand.

### Significance

Unlike other data-generating algorithms for offline RL, HIPODE is decoupled from the downstream offline RL policy, which allows it to be plugged into any existing offline RL algorithm.

**Weaknesses:**

1. In Section 3, the paper compares the results of the downstream offline RL algorithm using two types of augmented data: high-diversity data and high-return data. The high-return data was generated from a well-trained offline policy with a higher return. Since it is well-known that offline RL algorithms tend to perform better on medium-expert datasets than medium datasets, I believe it is evident that the downstream offline RL algorithm performs better with high-return augmented data. The two augmented data should have a similar maximum return value for a fair comparison.

2. The paper compares the performance of HIPODE with CABI. HIPODE and CABI share common aspects, such as using forward and inverse dynamics models. A careful analysis of how they differ from each other would be helpful for the readers to understand the novelty of HIPODE.

### MInor comments

1. p.5 $ \arg\max_{\tilde{s}'_{\text{cand}}}  V(\tilde{s}' _{\text{cand}}), \tilde{s}' _{\text{cand}}\in \tilde{S}' _{\text{cand}} \rightarrow \arg\max _{\tilde{s}' _{\text{cand}}\in \tilde{S}' _{\text{cand}}} V(\tilde{s}' _{\text{cand}})$

2. Section 5.1 Evaluation settings: Re-cite CABI, MOPO, and COMBO for clarity.

**Questions:**

Please refer to the **Weaknesses** section.

---

> ### Author Response · Authors · 2023-11-18
>
> Thanks to the reviewer for such a careful review!
>
> ## The two augmented data should have a similar maximum return value for a fair comparison.
>
> In section 3, we would like to explore whether augmenting high-reward (but low-diversity) data or high-diversity (low-reward) data is more effective. We agree that it is evident that the downstream offline RL algorithm performs better with high-return augmented data. However,  in offline setting, augmentation for a specific dataset does not allow generating data that is too far from the dataset, e.g., it is not possible to generate expert data when only the medium dataset is known. Intuitively, diverse data can expand the dataset coverage and thus improve the downstream algorithm generalization, which may also be beneficial to the dataset. So we think it makes sense to investigate which of these two types of data is better in data augmentation for offline RL. In fact, in the problem of data augmentation in offline RL for specific datasets, there is no previous work that explores exactly what characteristics of data are better to use.
>
> In summary, section 3 provides a fair comparison of the two data types and suggests that generating high-return data for a specific dataset may improve offline RL more.
>
> ##  How HIPODE and CABI differ from each other?
>
> 1. CABI does not consider return, but HIPODE focuses on generating high-return data.
>
> 2. CABI use a bi-direction generation, but HIPODE only generates transition in the forward direction.
>
> 3. CABI generates s' by (s,a), in both forward or reverse rolling. HIPODE generates (s,s') and then uses a reverse dynamic model to generate action.
>
> 4. HIPODE uses negative sampling to avoid OOD state to ensure data authenticity, while CABI uses double check mechanism.
>
> ## Minor issues
> We will change them in the next version paper.

---

> > ### Comment · Reviewer_Pxud · 2023-11-22
> >
> > Thank you for your response.

---

### Official Review · Reviewer_q6Nk · 2023-10-31

**Soundness:** 3 good
**Presentation:** 2 fair
**Contribution:** 3 good
**Rating:** 5
**Confidence:** 4

**Summary:**

This paper proposes HIPODE, a novel data augmentation technique for offline reinforcement learning that generates high-quality (i.e. high-return) augmented data in a policy-independent fashion. At a high level, HIPODE generates augmented transitions with high value estimates near the support of the offline dataset. Empirically HIPODE is competitive with other augmentation strategies and improves performance across several offline RL algorithms.

**Strengths:**

1. Many prior data augmentation works have demonstrated the effectiveness of different augmentation strategies and frameworks, but the question of what augmented data *should* be generated is much less studied (I expand on this in the Weaknesses section). Thus, this paper is novel and quite timely.

2. Empirical analysis is thorough: a variety of tasks and baselines are considered.

3. The proposed algorithm is intuitive, indeed generates high-value augmented data (i.e. it accomplishes what the papers claims it should accomplish, shown in Figure 4)

**Weaknesses:**

1. My primary concern relates to the first contribution, “Our findings indicate that high-return data, as opposed to noisy data with high diversity, benefits downstream offline policy learning performance more.” I think the community implicitly understands that offline RL algorithms perform with high-quality “expert” data, though few works explicitly state this claim. I think it’s absolutely a point worth discussing, though I wouldn’t consider it a core contribution. The claim itself is also difficult to assert, since prior and current works show that the story isn't so clear-cut. For instance:

* Kumar et. al [1] discuss how offline RL algorithms benefit from noisy expert data.
* Yarats et. al [2] show that vanilla off-policy RL algorithms can outperform state-of-the-art offline RL algorithms when given highly diverse data.
* Corrado et. al [3] introduce a framework for generating expert-quality augmented data that outperforms random data augmentation frameworks.
 * Corrado et. al [4] show that increasing an agent's state-action diversity via augmentation often yields more improvement than increasing the amount of reward signal an agent receives via augmentation.
* MoCoDA [5] is a data augmentation framework that enables users to directly control the distribution of augmented data generated. In particular, the user a can ensure task-relevant data is generated. This work also outperforms random data augmentation frameworks (including its predecessor CoDA [6])

I suggest rephrase section 3 as more of a didactic example illustrating how high-value augmented data can be more useful than diverse data. I also suggest including some or all of these works in the paper's related work section -- particularly [3] and [5], since both of these works focus on generating high-value augmented data.

1. My second concern is that the method seems to focus on generating high-value augmented data with low diversity (or low state-action coverage), but as mentioned above, data diversity is also quite important to the success of data augmentation. Thus, HIPODE may have limited applicability. I don't consider this a huge drawback though; in principle, one could use HIPODE to generate high-value augmented data along with an augmentation strategy that generates highly diverse (and potentially low-value) augmented data. It would be interesting to run experiments with "Original + Return + Diversity X" augmented data (using the naming convention of Table 1).

1. I'm skeptical about using learned model to generate rewards. Offline RL requires access to a reward function, so why not simply label augmented transitions with their true reward? Generating augmented rewards seems like an unnecessary source of variability.

1. Intuitively, I would expect HIPODE to offer the largest performance boost with random D4RL datasets (datasets which contain little to no high-value data), but its difficult to assess whether this intuition is true given the current presentation of results in Tables 2 and 3. I suggest grouping table rows by dataset type (random, medium, expert) and discussing general performance trends across dataset types.

2. The paper should include significance test for results reported in Table 1 and Table 2. (e.g. paired-t test at a 95% confidence level). For instance, in Table 1, the “Original + Return” returns for “halfcheetah-m-r” look very similar to returns for the other datasets.

1. Minor suggestion: the empirical section may flow a bit better if you first show HIPODE indeed generates high-value augmented data and *then* show that HIPODE improves performance.

I currently lean to reject, but I do like the ideas presented in this work.

[1] Kumar et. al. When Should We Prefer Offline Reinforcement Learning Over Behavioral Cloning? ICLR 2022

[2] Yarats et. al.  Don’t Change the Algorithm, Change the Data: Exploratory Data for Offline Reinforcement Learning. arXiv:2201.13425

[3] Corrado et. al. Guided Data Augmentation for Offline Reinforcement Learning and Imitation Learning. arxiv:2310.18247

[4] Corrado & Hanna. Understanding when Dynamics-Invariant Data Augmentations Benefit Model-Free Reinforcement Learning Updates. arxiv:2310.17786

[5] MoCoDA: Model-based Counterfactual Data Augmentation. Pitis et. al, NeurIPS 2022.

[6] Counterfactual Data Augmentation using Locally Factored Dynamics. Pitis et. al, NeurIPS 2020.

**Questions:**

1. The notion of “policy-decoupled” data augmentation is not clearly defined. Could the authors please clarify the difference between a policy-coupled a policy-decoupled data augmentation strategy?

1. The purpose of Figure 2 is unclear to me. The core claim of section 3 is that high-quality data is better than random data, and this claim is supported by Table 1 (somewhat--see next point). Figure 2 seems to simply show that high-quality data has the highest reward (which is obvious, by definition). What’s the purpose of t-sne here?

1. HIPODE generates augmented data that remains close to the support of the offline dataset. S4RL essentially adds small random perturbations to data and thus also generates data close to dataset’s support. Would it be correct to say that HIPDOE is much more careful version of S4RL?

1. Could HIPODE in principle be used for online RL?

---

> ### Author Response · Authors · 2023-11-18
>
> Thank you to the reviewer for reading the paper so carefully and making valuable suggestions!
>
> ##   The community implicitly understands that offline RL algorithms perform with high-quality “expert” data.
>
> We have taken note of the commendable works highlighted by the reviewer. Diverse data can intuitively expand dataset coverage, thereby enhancing downstream algorithmic generalization. Previous research by [1] demonstrated that noisy ground-truth data benefits Offline RL, but their setting differed from data augmentation for specific datasets. Data augmentation for specific datasets in an offline setting accounts for unique characteristics. Performance degradation may occur when there is an excessive amount of synthetic data or excessive noise, due to the unavailability of ground-truth access.
>
> In order to maintain fairness and emulate an offline data augmentation setting, we use a low synthetic portion. We  also use an offline RL policy for the `Quality' augmentation policy that avoids OOD actions. To our knowledge, no previous research has examined the effectiveness of specific data augmentation characteristics in improving offline RL.
>
> In summary, section 3 provides a fair comparison of the two data types and suggests that generating high-return data for a specific dataset may improve offline RL more.
>
> [1]  Kumar et. al. When Should We Prefer Offline Reinforcement Learning Over Behavioral Cloning? ICLR 2022
>
> ## Add related works.
>
> Many thanks to the reviewers for presenting such relevant and high-quality works, and we will include them in the related work and discuss them in the next version. We think this will greatly enhance the knowledge about dataset aspects in offline RL.
>
> ## What about augmenting with both high-quality and high-diversity data?
>
> This is really a valuable idea. We're experimenting with it and trying to put it up before the end of rebuttal.
>
> ## Offline RL requires access to a reward function.
>
> Our offline problem setting is only knowing the dataset and nothing else. The reward function can be seen as part of the environment which is not accessible. to the best of our knowledge, offline RL algorithms such as CQL or TD3BC are also set up in this way.
>
> ##   Grouping table rows by dataset type.
>
> We believe we have actually done that. Our next version paper will add on gain information for each type of task. We agree with the reviewer's intuition and have found that the random and medium-random datasets have larger boosts. This is due to the greater lack of high quality data in an already diverse dataset like a random dataset.
>
>  ##   Definition of  “policy-decoupled” data augmentation is not clear.
>
> The data distribution generated by a policy-coupled (dependent) method is related to the downstream policy, while policy-decoupled do not, as shown by the cross in Figure 1 of the paper.  For instance, RAMBO, a policy-coupled algorithm, generates synthetic data with a low value of $V^{\pi}$ (s') by employing the policy $\pi$ in the policy iteration. BooT, another policy-coupled algorithm, produces data that changes dynamically while the transformer model is being trained.
>
> The drawbacks of the policy-dependent (coupled) approach are that the data generation process and the policy iteration process must alternate, and the data adapted to one policy cannot be used for another. In contrast, the policy-decoupled approach allows for the reuse of synthetic data for different downstream algorithms. Furthermore, dividing the data generation process from the policy iteration process downstream enhances this module's flexibility for merging with other processes. For instance, human experience may be utilized to adjust augmentation data at low cost, without policy training iteration.
>
>  ## The purpose of Figure 2 is unclear.
>
> I'm very sorry we didn't make this clear in the paper. Figure 2 illustrates that the high-return(quality) policy indeed generates data with higher return, so that it can be sure that the better performance of the downstream offline policy do come from higher-return data.
>
> ## HIPDOE is a much more careful version of S4RL.
>
> We agree with the reviewer! However, HIPODE and S4RL has a lot of differences.  S4RL
>
> 1.Utilizes the augmented states for encouraging smoothness.
>
> 2.It is implemented by modifying the standard bellman error.
>
> 3.S4RL only augments states.
>
> 4.S4RL don't consider the return (value) of the noisy states.
>
> In contrast, HIPODE do not make any modifications to the downstream offline RL algorithm. In the other hand, HIPODE generates the full transition data with high return.
>
> ##  Could HIPODE in principle be used for online RL？
>
> This is really an intriguing idea！Thanks to the reviewer! We will try to use HIPODE to augment the replay buffer of a online algorithm and see if it works better than existing model-based methods like MBPO. We will try our best to obtain some results before the rebuttal ends.

---

> > ### Author Response · Authors · 2023-11-18
> >
> > ## Significance test for results
> >
> > We have counted the standard deviations of both Tabel1, Table 2 and Table 3 in the paper as follows.
> >
> > Table 1:
> >
> > | task name       | augmenting type         | td3bc        | cql           | iql          |
> > | --------------- | ----------------------- | ------------ | ------------- | ------------ |
> > | halfcheetah-r   | original                | 12.8$\pm$1.9 | 17.0$\pm$9.3  | 16.5$\pm$0.6 |
> > |                 | original+diversity 0.01 | 12.1$\pm$0.6 | 2.0$\pm$0.2   | 11.2$\pm$4.0 |
> > |                 | original+diversity 0.1  | 12.0$\pm$0.5 | 16.1$\pm$5.5  | 14.3$\pm$3.0 |
> > |                 | original+diversity 1.0  | 9.2$\pm$0.7  | 3.0$\pm$0.4   | 12.6$\pm$6.2 |
> > |                 | original+return         | 25.8$\pm$0.0 | 23.8$\pm$0.3  | 21.2$\pm$4.7 |
> > | halfcheetah-m-r | original                | 43.3$\pm$0.6 | 42.5$\pm$0.9  | 41.1$\pm$1.1 |
> > |                 | original+diversity 0.01 | 44.6$\pm$0.6 | 38.4$\pm$6.7  | 43.3$\pm$0.3 |
> > |                 | original+diversity 0.1  | 44.3$\pm$0.3 | 1.8$\pm$0.1   | 44.0$\pm$0.2 |
> > |                 | original+diversity 1.0  | 41.8$\pm$0.7 | 26.6$\pm$19.2 | 40.1$\pm$0.3 |
> > |                 | original+return         | 46.8$\pm$0.1 | 52.6$\pm$0.1  | 44.6$\pm$0.2 |
> >
> > Score of 12 Mujoco-v0 datasets of Tabel 2:
> >
> > | Task Name       | CQL+ HIPODE     | CQL+ CABI        | CQL           | TD3BC+ HIPODE    | TD3BC+ CABI    | TD3BC         |
> > | --------------- | --------------- | ---------------- | ------------- | ---------------- | -------------- | ------------- |
> > | halfcheetah-m-e | $99.5\pm{4.5}$  | $101.0$ $\pm$1.2 | 94.8$\pm$3.9  | $102.6\pm{2.2}$  | 94.6$\pm$8.5   | 98.0$\pm$3.1  |
> > | hopper-m-e      | $112.1\pm{0.1}$ | 112.0 $\pm$0.1   | 111.9$\pm$0.1 | $112.4\pm{0.3}$  | 102.8$\pm$16.0 | 112.0$\pm$0.1 |
> > | walker2d-m-e    | $94.0\pm{4.0}$  | 92.3$\pm$13.8    | 70.3$\pm$5.7  | $105.3\pm{4.0} $ | 99.6$\pm$6.4   | 105.4$\pm$3.9 |
> > | halfcheetah-m-r | $46.0\pm{0.1}$  | 42.8$\pm$2.0     | 42.5$\pm$0.9  | $44.0\pm{0.7} $  | 43.4$\pm$0.5   | 43.3$\pm$0.6  |
> > | hopper-m-r      | $34.7\pm{2.9}$  | 29.7$\pm$2.7     | 28.2$\pm$1.2  | $36.2\pm{0.8} $  | 32.7$\pm$2.3   | 32.7$\pm$0.5  |
> > | walker2d-m-r    | $21.7\pm{6.5}$  | 12.7$\pm$6.7     | 5.1$\pm$3.6   | $36.4\pm{11.1}$  | 37.7$\pm$15.4  | 19.6$\pm$8.6  |
> > | halfcheetah-m   | $39.3\pm{0.2}$  | 36.7$\pm$0.2     | 39.2$\pm$0.4  | $43.7\pm{0.5}$   | 43.0$\pm$0.3   | 43.7$\pm$0.4  |
> > | hopper-m        | $30.4\pm{0.9}$  | 30.5$\pm$0.3     | 30.3$\pm$0.7  | $99.9\pm{0.3} $  | 99.7$\pm$0.2   | 99.9$\pm$0.9  |
> > | walker2d-m      | $75.7\pm{5.1}$  | 46.8$\pm$16.1    | 66.9$\pm$9.3  | $80.1\pm{0.7} $  | 80.3$\pm$2.0   | 79.7$\pm$1.8  |
> > | halfcheetah-r   | $23.1\pm{1.5}$  | 2.8$\pm$0.9      | 17.0$\pm$5.7  | $15.5\pm{0.9} $  | 12.5$\pm$1.5   | 12.8$\pm$1.9  |
> > | hopper-r        | $10.4\pm{0.1}$  | 10.2$\pm$0.1     | 10.4$\pm$0.1  | $10.9\pm{0.2} $  | 10.9$\pm$0.0   | 10.9$\pm$0.2  |
> > | walker2d-r      | $10.5\pm{0.6}$  | -0.1$\pm$0.0     | 1.7$\pm$0.9   | $6.6\pm{1.0}  $  | 3.0$\pm$0.7    | 0.37$\pm$0.1  |
> > | Average         | 49.8            | 43.1             | 43.2          | 57.8             | 55.0           | 54.9          |
> >
> > Table 3:
> >
> > | Task Name       | Repeat+TD3BC  | NoV+TD3BC     | HIPODE+TD3BC  | TD3BC         |
> > | --------------- | ------------- | ------------- | ------------- | ------------- |
> > | halfcheetah-m-e | 97.4$\pm$3.6  | 99.1$\pm$4.2  | 102.6$\pm$2.2 | 98.0$\pm$3.7  |
> > | hopper-m-e      | 111.9$\pm$0.5 | 110.5$\pm$3.5 | 112.4$\pm$0.3 | 112$\pm$0.1   |
> > | walker2d-m-e    | 38.0$\pm$64.4 | 102.5$\pm$5.6 | 105.3$\pm$4.0 | 105.4$\pm$3.9 |
> > | halfcheetah-m-r | 42.5$\pm$1.6  | 44.0$\pm$0.2  | 44.0$\pm$0.7  | 43.3$\pm$0.6  |
> > | hopper-m-r      | 36.5$\pm$4.2  | 36.2$\pm$8.0  | 36.2$\pm$0.8  | 32.7$\pm$0.9  |
> > | walker2d-m-r    | 18.0$\pm$14.4 | 30.0$\pm$4.1  | 36.4$\pm$11.1 | 19.6$\pm$8.6  |
> > | halfcheetah-m   | 43.6$\pm$0.8  | 43.1$\pm$0.5  | 43.7$\pm$0.5  | 43.7$\pm$0.4  |
> > | hopper-m        | 99.8$\pm$0.1  | 99.7$\pm$0.3  | 99.9$\pm$0.3  | 99.9$\pm$0.5  |
> > | walker2d-m      | 79.5$\pm$2.7  | 79.7$\pm$3.1  | 80.1$\pm$0.7  | 79.7$\pm$1.8  |
> > | halfcheetah-r   | 11.7$\pm$0.3  | 13.1$\pm$1.4  | 15.5$\pm$0.9  | 12.8$\pm$1.9  |
> > | hopper-r        | 11.1$\pm$0.0  | 10.9$\pm$0.1  | 10.9$\pm$0.2  | 10.9$\pm$0.2  |
> > | walker2d-r      | 1.9$\pm$0.4   | 2.1$\pm$0.7   | 6.6$\pm$1.0   | 0.4$\pm$0.1   |
> > | halfcheetah-e   | 104.2$\pm$3.3 | 105.3$\pm$0.7 | 106.5$\pm$0.5 | 107.0$\pm$0.9 |
> > | hopper-e        | 112.5$\pm$0.5 | 112.3$\pm$0.2 | 112.3$\pm$0.5 | 107.3$\pm$8.7 |
> > | walker2d-e      | 78.1$\pm$48.5 | 104.9$\pm$5.8 | 106.6$\pm$5.2 | 98.7$\pm$6.0  |
> > | Average         | 59.1$\pm$9.7  | 66.2$\pm$2.6  | 67.9$\pm$1.9  | 64.7$\pm$2.6  |

---

> > ### Author Response · Authors · 2023-11-21
> >
> > ## What about augmenting with both high-quality and high-diversity data?
> >
> > This is really a valuable idea. We have obtained results on TD3BC and IQL about this. The results are as follows.
> >
> > |                 | dataset          | TD3BC        | IQL          |
> > | --------------- | ---------------- | ------------ | ------------ |
> > | halfcheetah-r   | Org              | 12.8$\pm$1.9 | 16.5$\pm$0.6 |
> > |                 | Org+Qua          | 25.8$\pm$0.0 | 21.2$\pm$4.7 |
> > |                 | Org+Qua+Div 0.01 | 25.9$\pm$0.3 | 19.8$\pm$3.1 |
> > |                 | Org+Qua+Div 0.1  | 25.8$\pm$0.2 | 19.9$\pm$3.9 |
> > |                 | Org+Qua+Div 1.0  | 25.5$\pm$0.7 | 25.0$\pm$0.3 |
> > | halfcheetah-m-r | Org              | 43.3$\pm$0.6 | 41.1$\pm$1.1 |
> > |                 | Org+Qua          | 46.8$\pm$0.1 | 44.6$\pm$0.2 |
> > |                 | Org+Qua+Div 0.01 | 46.5$\pm$0.1 | 44.8$\pm$0.1 |
> > |                 | Org+Qua+Div 0.1  | 46.7$\pm$0.2 | 44.9$\pm$0.1 |
> > |                 | Org+Qua+Div 1.0  | 45.3$\pm$0.5 | 43.6$\pm$0.1 |
> >
> > In the present experiment, the quantities of high-quality and diversity data were both half of the original data. The experimental results show that adding only high quality data significantly improves the performance compared to the original data. And then after continuing to add diversity data, the performance is almost unchanged.  This suggests that high quality data plays a greater role in data augmentation to improve the offline RL performance, when all data is completely realistic .

---

> > > ### Comment · Reviewer_q6Nk · 2023-11-22
> > >
> > > Thank you for the clarifications! In particular, the following points are now clearer to me:
> > > * Why HIPDOE uses a learned model to generate rewards.
> > > * The definition of "policy-decoupled" data augmentation
> > > * The differences between HIPDOE and S4RL
> > > * The purpose of Fig 2.
> > >
> > > I'd like to clarify my comment regarding how "the community implicitly understands that offline RL algorithms perform with high-quality 'expert' data." My view is that understanding how high-return data is generally more useful than noisy diverse data is the *motivation* for developing a data augmentation technique like HIPDOE, but this understanding is not a contribution in itself. I feel that the question of "should I generate high-return data or highly-diverse data" in Section 3 should be phrased as a point of discussion instead -- something like "Here are the advantages of high-return over high-diversity data, and we're going to illustrate this empirically." Overall, I am a bit less concerned about this point, and believe minor revisions can reframe this narrative.
> > >
> > > While the purpose of Fig 2 is now clear, I maintain my original comment: Fig 2 simply shows that high-return (quality) policy indeed generates data with higher return, but this seems obvious from the provided description of "policy with high-return." I think this figure adds marginal value to Section 3 and runs a risk of distracting readers like myself. I suggest moving this figure to an appendix. This is a minor comment though, and I'm glad I now understand the authors' intentions here.
> > >
> > > I'm personally curious to see how HIPDOE performs if you use the true reward for augmented data, since we do know the reward function in many offline settings. Intuitively, I expect HIPDOE to perform slightly better with true rewards. When the paper is eventually published (if not here, then somewhere else), this ablation would be helpful in understanding the degree to which these augmented rewards affect learning. To be clear, I'm not requesting the authors to run these experiments on the last day of discussion :)
> > >
> > > I ended up reading over the submission again (with revisions), and I'm glad to see the Kumar et. al and Yarats et. al references mentioned in the first paragraph on page 2. These papers are especially relevant to the question of "high return vs high diversity" data (particularly Yarats et. al).
> > >
> > > I also noticed a typo after looking at the paper again:
> > > * Page 2, 2nd paragraph: "while also ensures the high-return" -> "while also **ensuring** the high-return"
> > >
> > > After reading all reviewer comments and author responses, I'm inclined to maintain my score. The updated results with +/- stdevs show marginal improvements, so it is unclear if HIPDOE is superior to CABI and MOPO as claimed. More concretely, In Table 2:
> > > * CQL+HIPDOE outperforms the other two CQL baselines with potential significance on 6/15 datasets of datasets (hopper-m-r, walker2d-m-r, walker2d-m, halfcheetah-r, walk2d-r, walker2d-e)
> > > * TD3BC+HIPDOE outperforms the other two TD3BC baselines with potential significance on 7/15 datasets (halfcheetah-m-e, hopper-m-e, hopper-m-r, half cheetah-r, walker2d-r, hopper-e, walker2d-e)
> > > * Either CQL+HIPDOE or TD3BC+HIPDOE outperforms all other methods with  with potential significance only in 3/15 datasets (halfcheetah-m-e, hopper-m-e, walker2d-e)
> > >
> > > I agree with reviewer Yzj2 that it can be difficult to compare performance with only 3-5 seeds; thus, I use the phrase "with potential significance" to describe results for which HIPDOE's +/-stdev does not overlap with other +/- stdevs. I observed similar trends looking at the additional experiments in the appendix; we see potentially significant improvements on a small fraction of datasets. Perhaps the authors might consider identifying datasets for which HIPDOE greatly improves performance, and focus empirical evaluation on these datasets (i.e. choose datasets that highlight the strengths of HIPDOE).
> > >
> > > While I still lean to reject, I want to again emphasize that I like concept of HIPDOE and how it starts discussion on we should be generating augmented data.

---

> > > > ### Author Response · Authors · 2023-11-23
> > > >
> > > > Many thanks to the reviewer for the detailed comments.
> > > >
> > > > ##   It can be difficult to compare performance with only 3-5 seeds
> > > >
> > > > Thanks to the reviewers for raising this issue.
> > > >
> > > > As indicated by the results, the std error on most datasets are quite small, which shows the relative stability of the proposed methods. Thus we believe the current results are significant to some extent. However, we agree that adding more seeds can further improve the significance of the results and we will add more seeds in the future.
> > > >
> > > > ##   Should focus empirical evaluation on the  greatly improved datasets.
> > > >
> > > > Thanks to the reviewers for this valuable suggestion, we have added Section C.7 to the Appendix and re-updated the paper. In this section, we analyze the impact of augmented data on downstream effect enhancement in terms of both data novelty and authority, on relatively large-enhancement datasets. For details, please refer to Appendix C.7 in the new version of our paper. We will also use HIPODE on more types of environments in the future to do further analyses.

---

### Official Review · Reviewer_97dv · 2023-10-31

**Soundness:** 3 good
**Presentation:** 3 good
**Contribution:** 2 fair
**Rating:** 6
**Confidence:** 4

**Summary:**

The impact of different types of augmented data on downstream offline reinforcement learning (ORL) algorithms is thoroughly examined by the researchers. Their findings reveal that high-quality data has a greater positive impact on the performance of downstream offline policy learning compared to noisy data with high diversity.

To address this, the authors propose HIPODE, a policy-decoupled data augmentation method designed specifically for offline reinforcement learning (ORL). HIPODE acts as a versatile plugin capable of augmenting high-quality synthetic data for any ORL algorithm while remaining independent of the downstream offline policy learning process. HIPODE is evaluated on D4RL benchmarks, and the authors demonstrate its enhancement of multiple model-free ORL baselines. Furthermore, HIPODE surpasses other policy-decoupled data augmentation approaches for ORL.

**Strengths:**

- It is important to have a data augmentation approach that is policy-agnostic.
- The idea is simple yet effective
- The paper is well-written and easy to follow.

**Weaknesses:**

- In the experiments, the authors only test the proposed approach in environments where the agent’s action has a cyclic pattern. For these tasks, the transition model could be relatively simple and easy to be learned.
- The experiments with only 3 random seeds are unreliable in terms of reinforcement learning.
- It seems that there is no theoretical statement for the proposed approach.

**Questions:**

- It would be great if the authors could also provide the average return of the offline dataset in Table 1 and Table 2.
- Is the proposed approach applicable to the environment with discrete action space? It would be great if the authors could provide more experiments on different environments, such as Atari.
- It is interesting that the proposed method significantly outperforms previous methods on walker. What’s the cause of it?

------
### Post rebuttal
I appreciate the authors' provide further response and explanation. I am willing to keep my original scores.

---

> ### Author Response · Authors · 2023-11-18
>
> Thank you to the reviewer for the insightful review!
>
> ## Only 3 seed.
>
> We evaluate HIPODE+IQL on the Mujoco-v2 datasets on 4 random seeds as shown in Table 8 in Appendix  C.2. We are also adding seed to the main experiment in Section 5.1 and will update the results.
>
> ## Only cyclic environments.
>
> We realized this as well. We also evaluate HIPODE+TD3BC on the -human datasets of Adroit, a  robotic manipulation environment and find that HIPODE also possesses gain effects, as shown in Table 9 in the Appendix.
>
> ##   No theoretical statement.
>
> We recognize that we are currently deficient in theory, and that some theoretical issues have yet to be proven. For example, Can high quality data augmentation really leads to a theoretical gain? We will continue to think about this in the future.
>
> ## More results on Atari environments.
>
> We will evaluate HIPODE on Atari. However, the Atari experiment is considered to be a bit challenging because the discrete actions can cause the generation of exactly the same data as the dataset. A recent work [1] tries data augmentation on Atari but finds that data augmentation can have negative impacts on downstream policy learning algorithms.
>
> [1] Hwang D, Park M, Lee J. Sample Generations for Reinforcement Learning via Diffusion Models[J]. 2023.
>
> ##  HIPODE significantly outperforms previous methods on walker2d.
>
> We have also observed this interesting result. A recent work [2] finds that a model learnt on an offline dataset can plan a precise trajectory with explicit OOD actions on the Walker2d environment. Thus, we believe that a dynamics model with a strong ability to generalize to novel synthetic data is easier to learn on Walker2d.
>
> HIPODE generates high-quality data that leads to substantial improvements across nearly all datasets. The varying degrees of improvement on different datasets may be attributed to factors such as the environment, dataset characteristics, and variations in downstream algorithms employed.
>
> [2]  Dong Z, Yuan Y, Hao J, et al. AlignDiff: Aligning Diverse Human Preferences via Behavior-Customisable Diffusion Model[J]. arXiv preprint arXiv:2310.02054, 2023.

---

### Official Review · Reviewer_Yzj2 · 2023-11-01

**Soundness:** 2 fair
**Presentation:** 3 good
**Contribution:** 1 poor
**Rating:** 3
**Confidence:** 5

**Summary:**

The paper introduces HIgh-return POlicy-DEcoupled (HIPODE), a novel data augmentation approach for Offline Reinforcement Learning (RL), designed to overcome the constraints of existing policy-dependent augmentation techniques. Unlike traditional methods that either add noise or rely on dynamics models yielding data of uncertain quality, HIPODE generates high-return synthetic data that is policy-agnostic, thus supporting a variety of Offline RL algorithms. The method uses negative sampling to identify and select states with high potential values from near the existing data distribution.

The paper's key contributions include the development of HIPODE as a universal plug-in capable of enhancing the performance of any Offline RL process, independent of the downstream policy. Through experiments on the D4RL benchmarks, the authors show that HIPODE improves upon several model-free offline RL baselines and policy-decoupled data augmentation methods.

**Strengths:**

- Clarity: The paper is clearly articulated, presenting the concepts and methodology in a manner that is easy for readers to understand.
- Empirical contribution: It presents empirical results that indicate HIPODE's potential for improvement over current data-augmentation methods and model-free offline RL baselines on D4RL benchmarks.

**Weaknesses:**

- Relevance of Dataset Quality Analysis: Section 3's analysis, while potentially informative, may have limited relevance to the paper's contributions. The use of the true environment for generating synthetic data is an idealized condition not typically accessible in practical Offline RL scenarios where learned dynamics models are employed. This idealization might overstate the effectiveness of high-return data augmentation as it does not account for inaccuracies that would be present when using a learned model for data generation. Moreover, the assertion that high-value states improve policy learning is somewhat tautological and may not be particularly insightful since it is well-understood that higher-quality datasets tend to yield better-performing policies.

- Strength of Empirical Results: The empirical results presented, particularly in Section 5.2, lack sufficient detail and statistical rigor to substantiate strong claims of improvement. The absence of confidence intervals or a deeper statistical analysis in Table 3 makes it difficult to discern the significance of the performance gains attributed to HIPODE. Without this, the evidence provided does not firmly establish HIPODE’s superiority over, e.g., the NoV method.

- Benchmarking Against Recent Advances: The paper does not include comparisons to some of the more recent and potentially more effective model-based methods such as RAMBO and ROMI, which are known to deliver strong performances across various data regimes. This omission raises questions about the competitiveness of HIPODE and whether the improvements it offers are indeed leading-edge when considering the full landscape of contemporary Offline RL approaches. The inclusion of these comparisons would be critical for a more comprehensive assessment of HIPODE's performance and its standing relative to the state-of-the-art.

**Questions:**

N/A

---

> ### Author Response · Authors · 2023-11-16
>
> We thank the reviewer for raising these issues！
>
> ###  Relevance of Dataset Quality Analysis
>
> We agree that the true environment is an idealized condition. What we want to explore in this section is whether it is more beneficial to augment high-return data or high diversity data. In order to do this, we need to control the authority of both types of data to be the same, so we consider making both types of augmentation data completely true to avoid the authority affecting the performance. We considered that the ground truth data cannot be used in the offline setting, so we set the augmentation strategy to an offline rl strategy, avoiding generating ood action, which is similar to the data augmentation that works in the offline setting.
>
> We also agree that high-value states improve policy learning is somewhat tautological. However, we believe that there may be the influence of other factors in the offline rl data. For example, intuitively, Diverse data expands the dataset coverage and thus improves generalization of the downstream policy, e.g., [1] concluded that noise data is beneficial for OFFLINE RL, but there setup is different from data augmentation for a specific dataset. Data augmentation for a specific dataset in the OFFLINE setting has some unique characteristics, e.g., Too large proportion of synthetic data and too much noise can be detrimental to policy learning. In fact, no previous work has explored what's better for data augmentation to specific datasets in offline setting.
>
> As a summarize, section3 compares the two kinds of data as fairly as possible while staying close to the data augmentation setting, and finds out that it is better to do data augmentation for a specific dataset with high-return data.
>
> ###  Strength of Empirical Results
>
>  We report the standard deviations in Table 3.
>
> | Task Name       | Repeat+TD3BC  | NoV+TD3BC     | HIPODE+TD3BC  | TD3BC         |
> | --------------- | ------------- | ------------- | ------------- | ------------- |
> | halfcheetah-m-e | 97.4$\pm$3.6  | 99.1$\pm$4.2  | 102.6$\pm$2.2 | 98.0$\pm$3.7  |
> | hopper-m-e      | 111.9$\pm$0.5 | 110.5$\pm$3.5 | 112.4$\pm$0.3 | 112$\pm$0.1   |
> | walker2d-m-e    | 38.0$\pm$64.4 | 102.5$\pm$5.6 | 105.3$\pm$4.0 | 105.4$\pm$3.9 |
> | halfcheetah-m-r | 42.5$\pm$1.6  | 44.0$\pm$0.2  | 44.0$\pm$0.7  | 43.3$\pm$0.6  |
> | hopper-m-r      | 36.5$\pm$4.2  | 36.2$\pm$8.0  | 36.2$\pm$0.8  | 32.7$\pm$0.9  |
> | walker2d-m-r    | 18.0$\pm$14.4 | 30.0$\pm$4.1  | 36.4$\pm$11.1 | 19.6$\pm$8.6  |
> | halfcheetah-m   | 43.6$\pm$0.8  | 43.1$\pm$0.5  | 43.7$\pm$0.5  | 43.7$\pm$0.4  |
> | hopper-m        | 99.8$\pm$0.1  | 99.7$\pm$0.3  | 99.9$\pm$0.3  | 99.9$\pm$0.5  |
> | walker2d-m      | 79.5$\pm$2.7  | 79.7$\pm$3.1  | 80.1$\pm$0.7  | 79.7$\pm$1.8  |
> | halfcheetah-r   | 11.7$\pm$0.3  | 13.1$\pm$1.4  | 15.5$\pm$0.9  | 12.8$\pm$1.9  |
> | hopper-r        | 11.1$\pm$0.0  | 10.9$\pm$0.1  | 10.9$\pm$0.2  | 10.9$\pm$0.2  |
> | walker2d-r      | 1.9$\pm$0.4   | 2.1$\pm$0.7   | 6.6$\pm$1.0   | 0.4$\pm$0.1   |
> | halfcheetah-e   | 104.2$\pm$3.3 | 105.3$\pm$0.7 | 106.5$\pm$0.5 | 107.0$\pm$0.9 |
> | hopper-e        | 112.5$\pm$0.5 | 112.3$\pm$0.2 | 112.3$\pm$0.5 | 107.3$\pm$8.7 |
> | walker2d-e      | 78.1$\pm$48.5 | 104.9$\pm$5.8 | 106.6$\pm$5.2 | 98.7$\pm$6.0  |
> | average         | 59.1$\pm$9.7  | 66.2$\pm$2.6  | 67.9$\pm$1.9  | 64.7$\pm$2.6  |
>
>
>
> ### Benchmarking Against Recent Advances (RAMBO and ROMI)
>
> We mention RAMBO and ROMI in our related work.
>
> We compare RAMBO in Table 8 in Appendix C.2, and we find that HIPODE+IQL is slightly less effective than RAMBO. Nonetheless, we want to say that policy-dependent approaches, including RAMBO, aim to enhance specific policies, HIPODE focuses on enhancing various offline RL algorithms at minimal cost, which may come at the expense of enhancing specific policies.
>
> We did not compare ROMI, following CABI, because Appendix C.1 of CABI mentions that `the ROMI codebase has a number of secondary components (https://github.com/wenzhe-li/romi), e.g., prioritising experience replay, modifying state information, targeting different domains with different rollout strategies for different domains, and assuming that the termination function is known'. HIPODE only use the original dataset to train models and does not assume access to any other knowledge about the environment.
>
> [1]  Kumar et. al. When Should We Prefer Offline Reinforcement Learning Over Behavioral Cloning? ICLR 2022

---

> > ### Author Response · Authors · 2023-11-23
> >
> > Dear reviewer, we have provided detailed responses, but have not yet hear back from you. We will appreciate it deeply if you could reply our rebuttal. We are sincerely looking forward to further discussions to address the reviewers concerns to our best. Thanks!

---

### Author Response · Authors · 2023-11-22

Many thanks to the reviewers for their careful review and valuable suggestions. We have uploaded a new version of the paper, in which we have added the analysis of the related papers mentioned by reviewer q6Nk, made the presentation of Section 3 clearer, modified the minor issues raised by reviewer Pxud, added more detailed information to Table 1, Table 2, Table 3 and added two seeds to HIPODE+CQL. We have marked the changes as highlighted in the new version. We would be happy to have a more in-depth discussion with the reviewers.

---

### Meta-Review · Area_Chair_vss3 · 2023-12-05

**Metareview:**

This paper proposes a data augmentation method for offline RL, which generates high-return synthetic data by selecting states near the dataset distribution with potentially high value, while staying decoupled to the policy to cater for diverse downstream tasks.

Reviewers found, and I generally agree, that the experimental results are not sufficiently convincing, and some motivations and claims require more clarification in the context of existing literature.

**Justification For Why Not Higher Score:**

Reviewers found, and I generally agree, that the experimental results are not sufficiently convincing, and some motivations and claims require more clarification in the context of existing literature.

**Justification For Why Not Lower Score:**

N/A

---

### Decision · Program_Chairs · 2024-01-16

Reject